# Cryo-EM structure of cortical microtubules from human parasite *Toxoplasma gondii* identifies their microtubule inner proteins

Xiangli Wang[1,4], Yong Fu[2,4], Wandy L. Beatty[2], Meisheng Ma[1], Alan Brown [3], L. David Sibley[2✉] & Rui Zhang [1✉]

In living cells, microtubules (MTs) play pleiotropic roles, which require very different mechanical properties. Unlike the dynamic MTs found in the cytoplasm of metazoan cells, the specialized cortical MTs from *Toxoplasma gondii*, a prevalent human pathogen, are extraordinarily stable and resistant to detergent and cold treatments. Using single-particle cryo-EM, we determine their ex vivo structure and identify three proteins (TrxL1, TrxL2 and SPM1) as bona fide microtubule inner proteins (MIPs). These three MIPs form a mesh on the luminal surface and simultaneously stabilize the tubulin lattice in both longitudinal and lateral directions. Consistent with previous observations, deletion of the identified MIPs compromises MT stability and integrity under challenges by chemical treatments. We also visualize a small molecule like density at the Taxol-binding site of β-tubulin. Our results provide the structural basis to understand the stability of cortical MTs and suggest an evolutionarily conserved mechanism of MT stabilization from the inside.

[1] Department of Biochemistry and Molecular Biophysics, Washington University in St. Louis, School of Medicine, St. Louis, MO, USA. [2] Department of Molecular Microbiology, Washington University in St. Louis, School of Medicine, St. Louis, MO, USA. [3] Department of Biological Chemistry and Molecular Pharmacology, Blavatnik Institute, Harvard Medical School, Boston, MA, USA. [4]These authors contributed equally: Xiangli Wang, Yong Fu. ✉email: sibley@wustl.edu; zhangrui@wustl.edu

*T*oxoplasma gondii is an obligate intracellular parasite of a wide variety of warm-blooded animals and an opportunistic pathogen of humans, infecting about one-third of the world's population[1]. T. gondii infection is typically asymptomatic in healthy individuals, but can cause serious diseases in immune-compromised or pregnant individuals, leading to blindness and birth defects[2]. In addition, studies in mice have suggested that T. gondii may permanently alter specific brain functions and behavioral responses[3]. T. gondii belongs to a large phylum Apicomplexa, which also includes the notorious human pathogen Plasmodium falciparum, one of several causative agents of malaria.

T. gondii is a model system for microtubule (MT) studies, as it contains at least five different types of tubulin-containing polymers, including the spindle (in replicating parasites), centrioles, the conoid, the intra-conoid MTs, and cortical MTs (also named subpellicular MTs)[4] (Fig. 1a). T. gondii has 22 cortical MTs that initiate from the apical polar ring, an atypical MT organizing center (MTOC)[5], extending backwards to cover two thirds of the length of the cell body. The cortical MTs are tightly associated with the inner membrane complex (IMC) and are spaced ~23 nm apart in T. gondii sporozoites (27 nm in Plasmodium sporozoites) as previously reported by a cryo-electron tomography (cryo-ET) study[6]. The IMC is located underneath the plasma membrane and is composed of the alveoli (a patchwork of flattened vesicles) and a highly stable network of intermediate filament (IF)-like proteins underneath the alveoli[7].

Cortical MTs in Apicomplexa are specialized MTs that form underneath the pellicle (the IMC-plasma membrane complex) and provide the cortical rigidity needed by the parasite to maintain its shape and withstand diverse environmental forces[8]. In addition to their presence in most motile stages of the Apicomplexa, cortical MTs are also found in many other pathogens such as Trypanosome species[9], as well as human platelets[10] and plants[11]. Unlike the dynamic MTs found in the cytoplasm of metazoan cells, cortical MTs in Apicomplexa and Trypanosomes are extraordinarily stable and resistant to detergent and cold treatments[12–14].

An emerging concept in the MT field is that MT-associated proteins (MAPs) bind and modify the mechanical properties of MTs[15]. To date, eight different MAPs for T. gondii cortical MTs have been reported, including SPM1/2, TrxL1/2, and TLAP1/2/3/4[16–18]. Low-resolution cryo-ET structures of cortical MTs from T. gondii and another related apicomplexan parasite, Plasmodium berghei, reveal the presence of luminal densities with 8-nm periodicity[19]. However, it is still unclear which set of proteins bind to the external or luminal surface of the MTs, and whether they bind directly to tubulin or via interactions with other MAPs.

Luminal particles have long been observed within the cytosolic MTs of neuronal cells by conventional electron microscopy[20] and recently by cryo-ET[21], although only one microtubule inner protein (MIP), MAP6, has been identified so far for mammalian cytosolic MTs[22]. A variety of MIPs densities have also been visualized in the doublet MTs of the axoneme and the triplet MTs of the basal bodies or centrioles[23,24]. Furthermore, 33 different MIPs have been recently identified within the doublet MT from Chlamydomonas reinhardtii[25]. Outside the cilia/basal body system, MIPs were also observed in MTs within the ventral disk of Giardia lamblia[26].

Here, to understand the molecular basis for the extraordinary stability of the cortical MTs of Apicomplexa, we determine the structure of cortical MTs from T. gondii using cryo-electron microscopy (cryo-EM) and single-particle analysis. This ex vivo structure of singlet MTs allows the identification of natively decorating MAPs (MIPs).

## Results

### Single-particle cryo-EM reconstruction of *T. gondii* cortical MTs.
Given our previous success in obtaining isolated ciliary doublet MTs for single-particle analysis[25], we initially attempted to separate the cortical MTs from the IMC using detergent treatment (see "Methods") that removes most of the membrane components. However, the cortical MTs remain associated with the IMC after treatment, probably due to their tight connection with the IF-like protein network underneath the alveoli[7]. Therefore, we proceeded to image the cortical MTs ex vivo after detergent treatment (Supplementary Fig. 1a, b) and process the cryo-EM data using single-particle analysis (Supplementary Table 1). Fortunately, the MT signal was sufficiently strong to overcome the background noise caused by the presence of extant cellular materials, as reflected by the well-resolved 2D class averages of the MT particles (Supplementary Fig. 1c). Eventually, we could obtain a cryo-EM structure of cortical MT with 8-nm periodicity at an overall resolution of 4.0 Å.

Using a divide-and-conquer strategy[25], we performed focused refinement on smaller sub-regions each containing only two adjacent protofilaments (Supplementary Fig. 1d, e), which further improved the resolution for the sub-regions to around 3.4 Å (Supplementary Fig. 1d). At this resolution, we could identify the protein components and build an atomic model for the entire complex.

Estimation of the defocus values for individual MT particles within a micrograph revealed two sets of defocus values that are 0.1-0.2 μm apart (Supplementary Fig. 2a), corresponding to the two sets of cortical MTs located at the 'front' and 'back' surfaces of the parasites (Fig. 1a). Given that cortical MTs are tightly associated with the IMCs, the much shorter thickness of the cell bodies (0.1–0.2 μm), compared with their width on the EM grid (typically 3–4 μm) (Supplementary Fig. 1a), indicates that the parasites are significantly flattened (Supplementary Fig. 2b), probably due to loss of osmotic pressure upon detergent treatment and/or the surface tension of the thin liquid film of the sample during cryo-freezing process. Notably, it is the cell body flattening that allowed us to perform single-particle analysis of the cortical MTs ex vivo, otherwise the whole cell would be too thick for the electron beam to pass through.

### T. gondii cortical MT has an asymmetric organization.
The cryo-EM structure of the T. gondii cortical MT revealed a 13-protofilament MT with 3-start helix (Fig. 1b–d) and a single seam, where α-tubulin interacts with β-tubulin. This structure resembles the predominant form of mammalian cytosolic MTs, although the cross-section of T. gondii cortical MT is slightly elliptical, with the seam located close to the apex of ellipse (Fig. 1b, c). In our structure, shelf-like protrusions with 8-nm periodicity are present at the luminal surface (Fig. 1d), corresponding to MIPs. These densities are also apparent in the 2D class averages of the MT particles (Supplementary Fig. 1c). All these MIPs bind near the intradimer interface between α- and β-tubulin, following the path of the 3-start helix of the MT (Fig. 1d). Interestingly, at higher isosurface threshold, the helical arrangement of these MIPs is not continuous due to the apparent absence of densities at three sites (Fig. 1b, black dashed circles). The 'emptiness' of sites x and y (actually partially occupied by another MIP, see later) may be related to the slightly higher curvature between neighboring protofilaments, whereas the lack of a MIP at site z (at the MT seam) can be explained by the altered binding site due to heterotypic tubulin lateral interactions.

Our single-particle structure is in good agreement with previous cryo-ET structures of cortical MTs from T. gondii and Plasmodium berghei[19], indicating that our structure likely

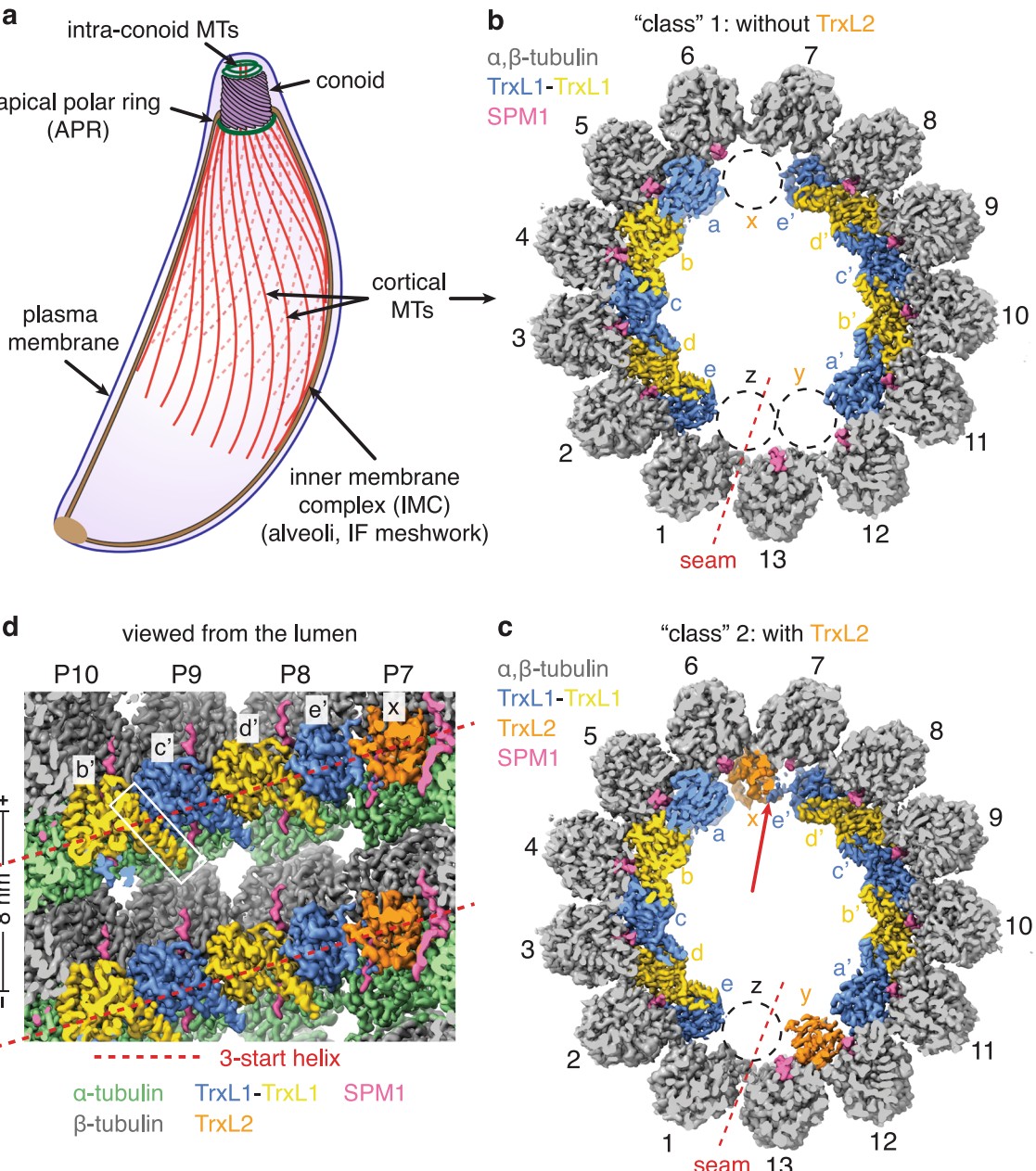

**Fig. 1 Overview of the cryo-EM structures of cortical MT from *T. gondii*. a** Cartoon diagram of *T. gondii* showing plasma membrane, inner membrane complex (IMC) (including the alveoli and an intermediate filament (IF)-like protein meshwork), the apical complex (a conoid and two intra-conoid MTs), the apical polar ring (APR) and 22 cortical MTs. **b–c** Cross section view of the cryo-EM structure of *T. gondii* cortical MT with (**b**) and without (**c**) TrxL2 at sites x and y. The structures show an asymmetric arrangement of the microtubule inner proteins (MIPs) including TrxL1 (gold or blue), TrxL2 (orange), and SPM1 (pink). An unoccupied site at the MT seam is labeled as z. Composite DeepEMhancer-sharpened maps (see "Methods") are used for visualization. The quotation marks on "class" 1 and "class" 2 indicate that the two composite maps only differ in sub-regions containing sites x and y. **d** The same as (**c**), but showing the luminal surface of the tubulin and MIPs. The N-terminal helix of TrxL1 (marked by white rectangle) inserts into a binding pocket of the neighboring TrxL1 molecule. The helical paths for the 3-start helix of the MT are indicated by dashed red lines.

maintains most, if not all, MIPs. However, we did not observe any defined protein densities at the external surface of the cortical MTs, even at very low isosurface threshold (Supplementary Fig. 3) (see "Discussion").

**Identification of TrxL1 and TrxL2 as MIPs.** The three 'empty' sites present in one helical turn divide the MIPs into two 'trains' on either side of the MT seam, each containing five copies of a globular protein (Fig. 1b). Using the cryo-EM densities, we identified this globular MIP to be TrxL1, which contains a

thioredoxin-like fold[17]. The N-terminal helix of each TrxL1 is inserted into a binding pocket of the neighboring TrxL1 molecule (Fig. 1d), except for the terminal ones at sites e and e' (Fig. 1b). In the absence of a 'downstream' molecule, the N-terminal helix is barely visible in the cryo-EM map (Figs. 1b, 2a), probably reflecting its intrinsic flexibility.

At lower isosurface threshold, two additional globular densities at sites x and y (Fig. 1c) become more apparent, each displaying a thioredoxin-like fold very similar to TrxL1 (Fig. 2a). Using focused 3D classification (see "Methods"), we could separate the bound and unbound states of these two protein densities at sites x

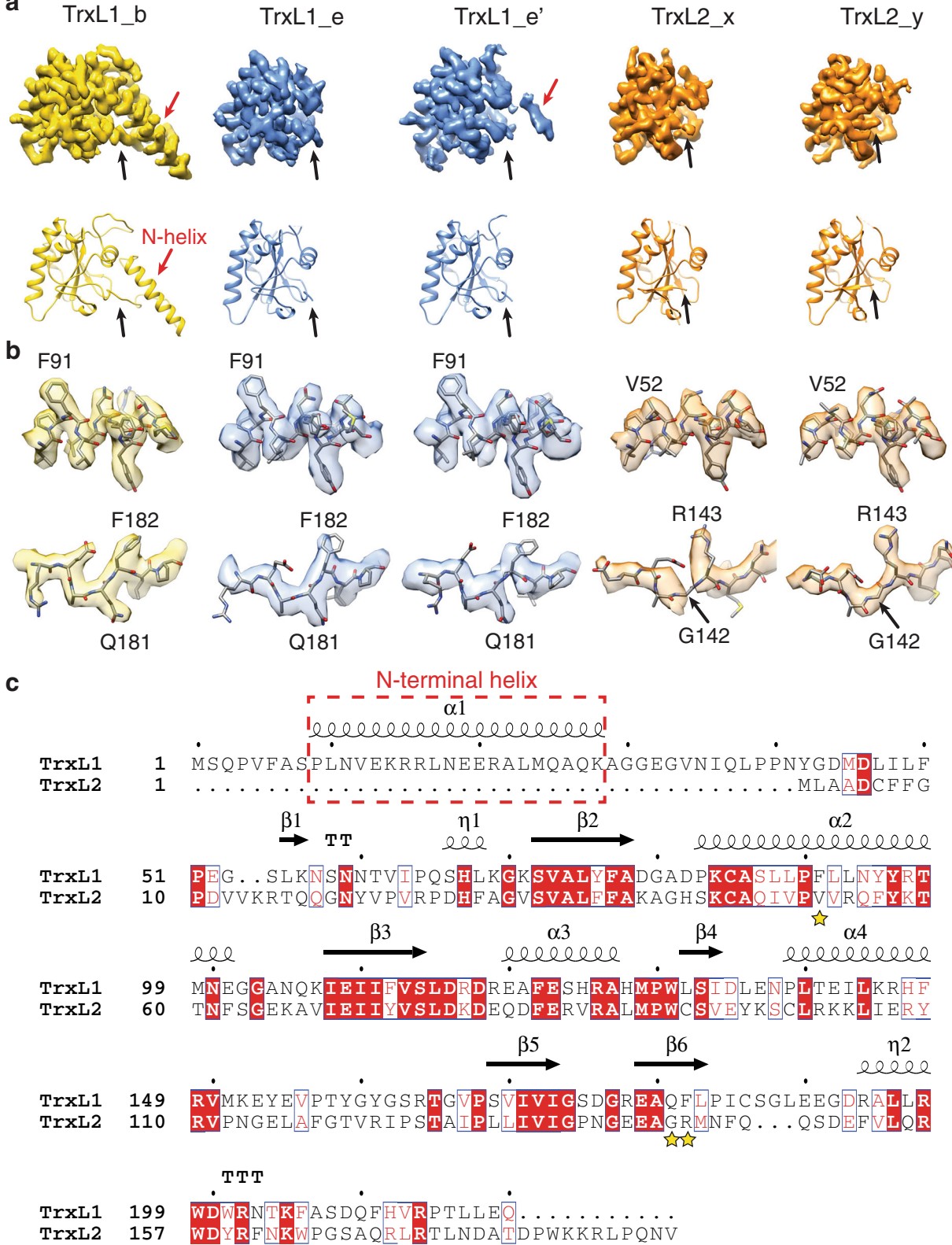

**Fig. 2 TrxL1 and TrxL2. a** Atomic models and segmented cryo-EM densities for TrxL1 (gold and blue) and TrxL2 (orange) at different locations as indicated in Fig. 1c. The black arrows highlight the distinct appearances of the N-termini of TrxL1 and TrxL2. **b** Side-chain densities of three residues (TrxL1: F91, Q181, and F182) that distinguish TrxL1 from TrxL2. DeepEMhancer-sharpened maps (see "Methods") are used for visualization. **c** Sequence alignment of TrxL1 and TrxL2. Secondary structure elements of TrxL1 are shown above the sequences. The three residues shown in (b) are marked by yellow stars.

and y (Fig. 1b, c). The well-resolved side-chains allowed us to confidently assign these two densities as belonging to TrxL2 (Fig. 2b), a paralog of TrxL1 that lacks the N-terminal helix[17] (Fig. 2c). Further data analysis indicated that the bound and unbound states of TrxL2 are not tightly correlated between sites x and y and are seemingly randomly distributed along the cortical MTs. This uneven distribution of TrxL2 may reflect the real scenario in living cells, or the dissociation of TrxL2 molecules during sample preparation for cryo-EM. It is also noteworthy that when TrxL2 is present at site x, the N-terminal helix of TrxL1 at site e' becomes visible (Fig. 1c, red arrow, and Fig. 2a).

**SPM1 is a filamentous MIP with Mn motifs**. At the interface between TrxL1/2 and tubulin, there is an elongated density on most protofilaments, except for protofilaments P1 and P7 (when TrxL2 are not present) (Figs. 1b, c, and 3a). This density is also weaker on P12 in the absence of TrxL2 (Supplementary Fig. 4, red arrows). The appearance of this density is strikingly similar to the MT-binding motif of FAP363, a MIP of the ciliary doublet MT from *Chlamydomonas*[25] (Fig. 3b). They both feature a short α-helix bound at the intra-dimer interface between α- and β-tubulin, with an apparent periodicity of 8 nm. Based on the side-chain densities for the consensus residues, we could confidently

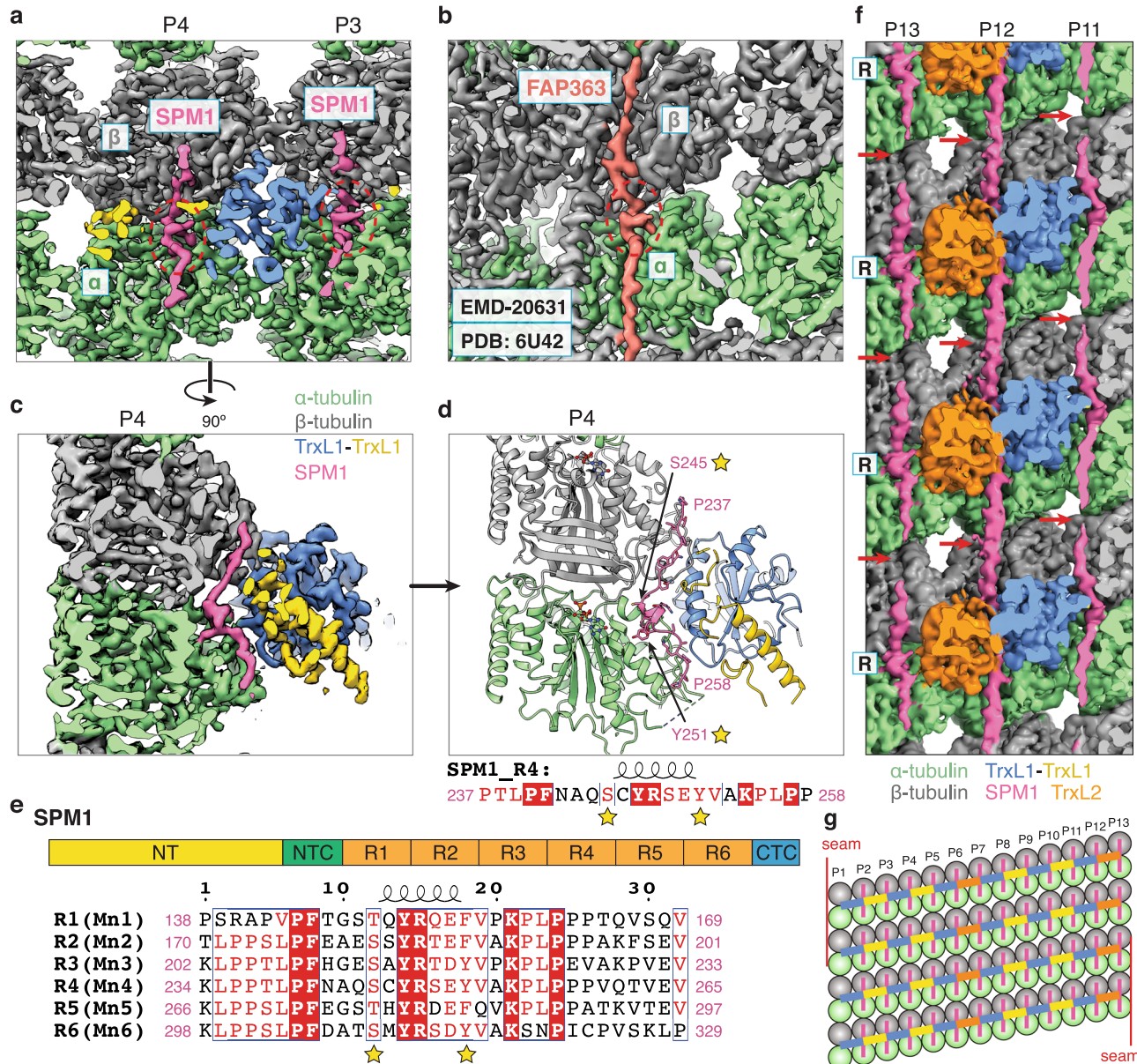

**Fig. 3 The interactions between SPM1, TrxL1/2, and tubulin. a** Cutaway view of the cryo-EM density showing a short α-helix (red dashed circle) of SPM1 (pink) bound at the intra-dimer interface between α- and β-tubulin (green and gray, respectively). **b** Cutaway view of the cryo-EM density of FAP363, a Mn-motif containing protein, bound to the *Chlamydomonas* ciliary doublet MT[25]. **c** 90-degree rotation of panel a, showing SPM1 is sandwiched between tubulin and TrxL1. **d** Same view as panel c, but only displaying the atomic models. The two conserved residues (characteristic of the Mn motif) of SPM1 are marked by yellow stars (same in panel e). We arbitrarily assigned the protein sequence of the SPM1 density to be R4 for the purpose of model building. **e** Above: schematic of the domain organization of SPM1. NT: N-terminal domain, NTC: N-terminal conserved domain, CTC: C-terminal conserved domain. Below: sequence alignment of the six internal repeats (R1–R6) of SPM1. **f** Luminal view of the MIP densities at protofilaments 11–13 (P11–13). Unsharpened cryo-EM density map is used to better visualize the connectivity of SPM1. The red arrows point to a region of SPM1 with weaker densities (except for the SPM1 on P12), presumably due to incoherent averaging of divergent residues between R1–R6. **g** Schematic of the MT lumen that is cut open and unfurled at the seam.

assign this elongated density as one of the six internal repeats (the R domain) of SPM1 (Fig. 3c-e), a protein that has been shown to decorate the entire length of cortical MTs by immunofluorescence microscopy[16].

In addition to the structural similarity, sequence alignment (Supplementary Fig. 5a) also indicated that SPM1 shares conserved motifs with FAP363 as well as a group of proteins named SAXO proteins (short for "stabilizer of axonemal MTs") that are present only in ciliated or flagellated organisms[27]. This consensus sequence, named the Mn motif, features two highly conserved T/S and F/Y residues respectively at positions 6 and 12 of the motif[27] (Fig. 3e and Supplementary Fig. 5a). The Mn motif is also present in the aforementioned human neuronal protein MAP6 (also called STOP protein)[28] (Supplementary Fig. 5a). In the atomic models of SPM1 and FAP363, these two conserved residues mark the start and end positions of the short helix that binds at the interface between α- and β-tubulin (Fig. 3d). Consistent with the suggested role of SPM1 in stabilizing the cortical MTs[16], it has been reported that human SAXO1 protein stabilizes MT structures within the centriole, basal body, and primary cilium[29], while the Mn module in MAP6 conveys resistance to both cold- and nocodazole-induced MT disassembly[30].

SPM1 contains six tandem copies of the R domain, each coincides with a Mn motif (Fig. 3e). Although we could confidently assign SPM1 based on the side-chain densities of the consensus residues, the resolved density of SPM1 appears to be an average of different Mn motifs. Unfortunately, we could not separate them in data processing due to the lack of globular domains to serve as fiducial markers for longer periodicity (e.g., 32 nm) (see further explanation in "Methods"). When displayed at lower isosurface threshold, the density of SPM1 forms a continuous or near-continuous linear density that runs along the protofilament (Fig. 3f), indicating that one SPM1 molecule only binds to one protofilament in the longitudinal direction (Supplementary Fig. 5b). Based on our modeling, all 32 residues of the R domain (Fig. 3e) are needed to fill the space of this continuous density and make a linear arrangement. In contrast, the six Mn motifs of FAP363 span two neighboring protofilaments (Supplementary Fig. 5b). It remains to be determined whether SPM1 proteins bind MTs using all six motifs in vivo.

**T. gondii tubulin isotypes**. The T. gondii genome encodes three α-tubulin and three β-tubulin isoforms[31,32]. The β-isoforms share high sequence identity (97%), while the three α-isoforms have quite distinct sequences (40 and 68% identity) (Supplementary Figs. 6 and 7), suggesting that they play different functions in living cells[4]. Mass spectrometry (M/S) analysis of our sample (Supplementary Data 1) revealed only one isoform of α-tubulin (583.m00022 (α1)) and two isoforms of β-tubulin (m57.00003 (β1) and 41.m00036 (β2)), which presumably are the major tubulin isoforms for T. gondii cortical MTs. At the current resolution, we cannot distinguish between β1 and β2 isoforms. These three identified isoforms (α1, β1 and β2) share high sequence identity to mammalian tubulin (Supplementary Figs. 6 and 7), and not surprisingly, the structure of T. gondii tubulin and their interactions are similar to their mammalian counterparts[33]. A more quantitative analysis of the tubulin lattice parameters[34] revealed subtle differences between T. gondii and mammalian MTs. The measured longitudinal spacing between T. gondii tubulin dimers is 82.4 Å, which is between mammalian GDP-MT (81.6 Å) and GMPCPP-MT (84.0 Å); while the measured lattice twist angle (-0.15 degree) is also distinct from the GDP-MT (0.1 degree) and GMPCPP-MT (0.2 degree).

**Unexplained density at the Taxol-binding site**. In our cryo-EM structure, we observed unexplained density within the Taxol-

binding pocket of every β-tubulin (Fig. 4a, b, red arrows). However, its size and shape are incompatible with Taxol (Fig. 4c, d) or other taxane drugs such as zampanolide or epothilone, which were not used throughout our parasite growth and sample preparation process. Mass spectrometry analysis of our sample did not detect any post-translational modifications (PTMs) on tubulin residues in the vicinity of this small density. We could also exclude the possibility of this density being part of the neighboring SPM1 for two reasons. (i) In our cryo-EM structure, SPM1 is not present on protofilaments P1 and P7 (without TrxL2) (Fig. 1b), but the small densities are, with an intensity level as strong as tubulin. (ii) SPM1 forms a continuous or near-continuous density along each protofilament (Fig. 3f), so the residues within the R domain cannot turn sideways into the taxol-binding pocket. It is also spatially impossible for this density to be missing residues of TrxL1/2.

Therefore, this density is most likely to be a small molecule. To date, all small molecules that bind to the Taxol-binding site stabilize the MTs[35]. In addition, within the highly stable ciliary doublet MT, some MIPs insert a segment into the Taxol-binding sites of β-tubulin[25,36]. Considering that no molecules with comparable size and shape were present in our experimental buffers, it is tempting to hypothesize that T. gondii endogenously produces a small molecule that stabilizes its own cortical MTs. This chemical strategy could serve as a fail-safe mechanism, which may explain the observations that T. gondii cortical MTs display normal morphology upon deletion of various MAPs/MIPs. Further study is needed to identify and characterize this small molecule.

**Probing the functions of individual MIPs**. Having identified TrxL1, TrxL2, and SPM1 as bona fide MIPs, we set to further investigate their physiological functions. We first generated single knockout strains of each MIP (Δtrxl1, Δtrxl2, Δspm1) and a double knockout strain (Δtrxl1Δtrxl2) using CRISPR-Cas9 techniques we have described previously[37] (Supplementary Fig. 8a–e). We then examined their fitness using an in vitro plaquing assay, which did not show any significant defects, suggesting these proteins are not individually required for tachyzoite growth (Fig. 5a, b). Next, we used CRISPR-Cas9 to generate endogenously C-terminal HA-tagged versions of each MIP (Fig. 5c, d, Supplementary Fig. 8f). As visualized by immunofluorescence assay (IFA), all three proteins localized along the full length of the cortical MTs but do not associate with the conoid (Fig. 5d), consistent with previous reports[16–18]. In all cases, deletion of individual MIPs did not substantially change the morphology of cortical MTs (Fig. 5d). We then examined the interdependence between these MIPs. Upon deletion of SPM1, TrxL1 became almost completely cytosolic (consistent with a previous observation[17]), and TrxL2 are also partially mislocalized to the cytosol (Fig. 5d). This dependence of TrxL1/2 on SPM1 agrees well with the spatial arrangement of SPM1, TrxL1/2, and tubulin revealed by our cryo-EM structure (Fig. 3c, d, f). It is also noteworthy that upon deletion of TrxL2, TrxL1 are partially misplaced (Fig. 5d). In contrast, both SPM1 and TrxL2 localize to cortical MTs independent of TrxL1 (Fig. 5d).

Previous studies suggested that MAPs (or MIPs) have remarkable functional redundancy, i.e., cortical MTs remain largely normal upon single knockout, even double or triple knockout, of any reported MAPs[16–18]. Therefore, to further understand their effects on MT stability, we challenged the cortical MTs in wild-type and knockout strains with three different chemical treatments (Fig. 6a, b). (i) Upon relatively mild detergent treatment with Triton X-100 after glycerol extraction, the cortical MTs in wild-type and Δtrxl1, Δtrxl2 strains display normal length distribution as visualized by negative-stain EM,

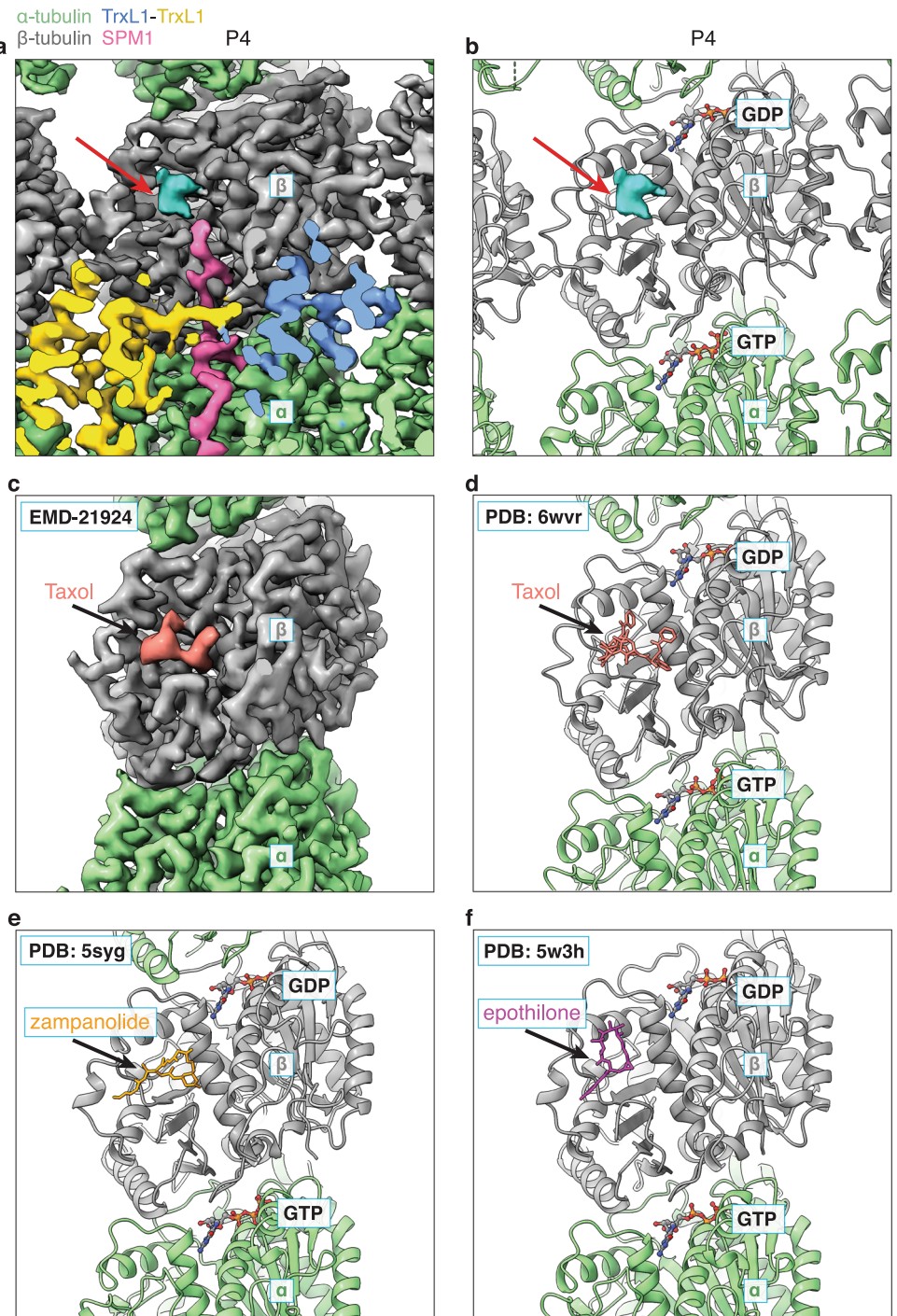

**Fig. 4 Density at the Taxol-binding site of β-tubulin. a** Close-up view of the cryo-EM density of one α,β-tubulin dimer at protofilament P4. Unexplained density (cyan, pointed by a red arrow) can be observed at the Taxol-binding site of β-tubulin (gray), but its size and shape are inconsistent with Taxol (**c, d**). **b** Same view as a, but showing the atomic models together with the unexplained density. Atomic models of TrxL1 and SPM1 are hidden for clarity. **c** Close-up view of the cryo-EM density of one α,β-tubulin dimer of taxol-stabilized bovine MT (EMD-21924)[66]. **d** Atomic model of the cryo-EM density shown in (**c**) (PDB 6WVR)[66]. **e** Atomic model of zampanolide-stabilized porcine MT (PDB 5SYG)[67]. **f** Atomic model of epothilone-stabilized yeast MT (PDB 5W3H)[68].

while no cortical MTs can be observed for Δ*spm1* or Δ*trxl1*Δ*trxl2*. (ii) Upon stronger detergent extraction condition with 1.5% cholic acid, again the cortical MTs within all the Δ*spm1* strain and most of the Δ*trxl1*Δ*trxl2* strain are completely disrupted. In the case of Δ*trxl1*, the length of cortical MTs are significantly reduced, while the conoid remains intact and the MTs near the apical end of the cell appear to be normal (Supplementary Fig. 9). (iii) Under a harsh condition of detergent treatment with 1.5%

cholic acid followed by trypsin digestion, the MT cytoskeleton within Δ*trxl1* and Δ*spm1* are completely disrupted. The cortical MTs within the wild-type and Δ*trxl2* maintain their normal length distribution but become disorganized. Trypsin digestion also leads to loss of the conoid, although the apical polar rings (Fig. 1a) are still present (Fig. 6 and Supplementary Fig. 9). In conclusion, our results indicate that both SPM1 and TrxL1 contribute to the stability of cortical MTs, with SPM1 being more

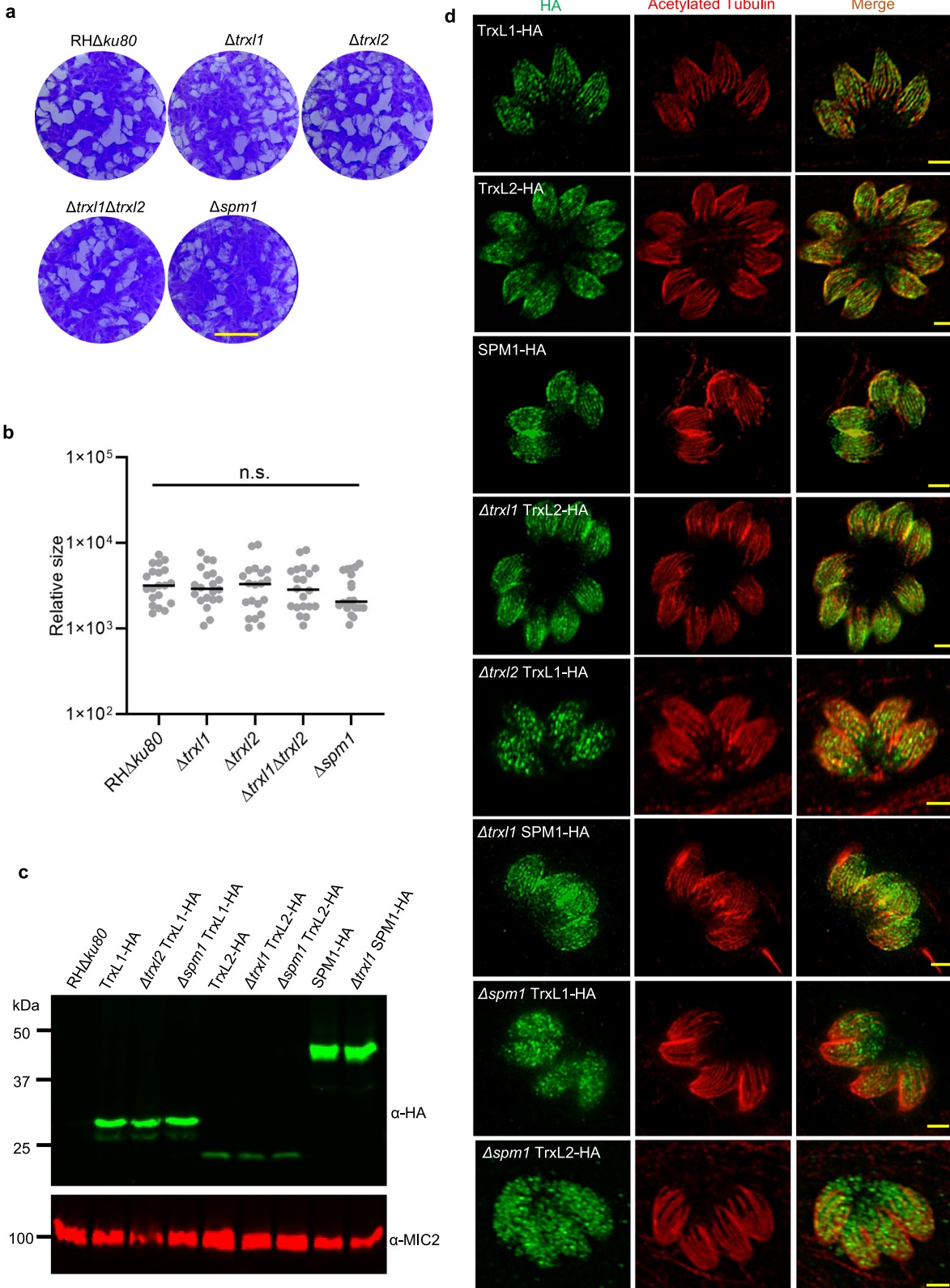

**Fig. 5 Knockout strains and HA-tagged strains of individual MIPs. a** Plaques formed by RHΔ*ku80* (wild type), Δ*trxl1*, Δ*trxl2*, Δ*trxl1*Δ*trxl2* and Δ*spm1* on human foreskin fibroblasts cells (HFF) monolayers infected with 200 parasites per monolayer. Scale bar = 5 mm. **b** Plaque area was measured by the pixels of each plaque and indicated as relative size. Plaque data represented 20 replicates for each parasite strain and was analyzed for statistical significance using one-way ANOVA. **c** Western blot analysis of different *T. gondii* strains expressing endogenously 3xHA (α-HA) tagged TrxL1, TrxL2, and SPM1. Tagged proteins were detected using anti-HA antibody (α-HA, green). MIC2 (α-MIC2, red) was used as a loading control. **d** Indirect immunofluorescence of different HA tagging strains . Tagged proteins were detected using anti-HA antibody (green) 24 h post-infection. Anti-acetylated tubulin antibody (red) was used to stain MTs for co-localization. Images were taken by laser scanning confocal microscope with Airyscan. Scale bar = 2 μm. Experiments in (**a**) and (**c**) were repeated twice with similar results. Each IFA image in (**d**) was selected from one of the three independent experiments with similar outcomes.

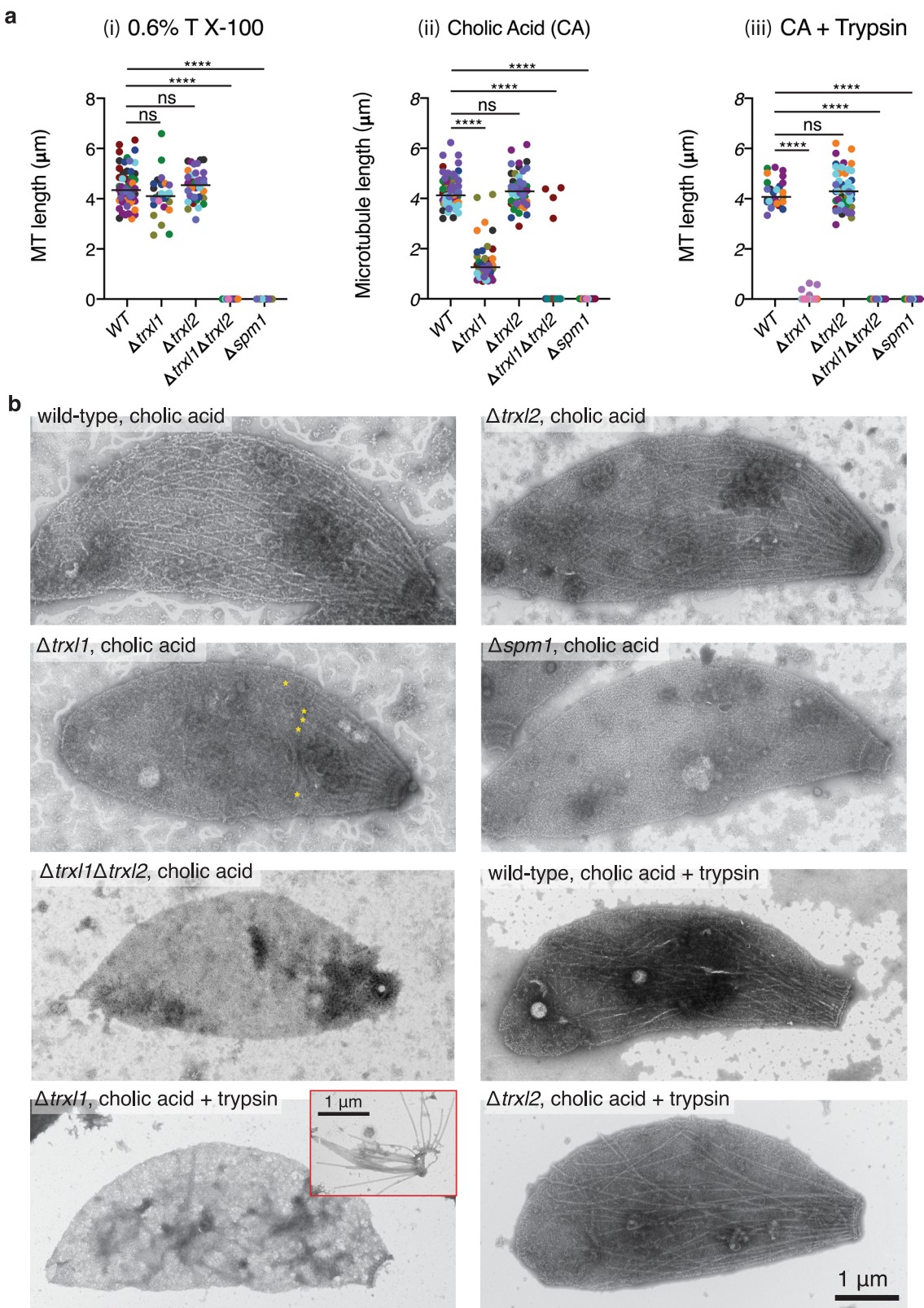

important, while the physiological function of TrxL2 is still unclear and awaits future investigation.

## Discussion
Our chemical treatment experiments and all previous literature indicate that SPM1 plays a major role in stabilizing the cortical MTs, a function that seems to be characteristic for Mn motif-containing proteins [29,30]. TrxL1 (and TrxL2) provides additional stability to the cortical MTs, by forming a shelf-like density in the lateral direction, which can strengthen the tubulin lateral interactions within the lattice, a MT-stabilization effect that cannot be offered by the longitudinally oriented SPM1 molecules (Fig. 3g). In the absence of SPM1, neither TrxL1 nor

**Fig. 6 Cortical microtubule lengths in wild-type and knockout strains upon chemical treatments. a** Plot of MT lengths of wild-type and four knockout strains (Δ*trxl1*, Δ*trxl2*, Δ*spm1*, and Δ*trxl1*Δ*trxl2* double knockout) upon three different chemical treatments: (i) glycerol extraction followed by detergent treatment with 0.6% Triton X-100 (58, 32 and 37 MTs were measured from 10 WT, 5 Δ*trxl1*, and 10 Δ*trxl2* parasites, respectively); (ii) detergent extraction with 1.5% cholic acid (68, 42, 59, and 4 MTs were measured from 10 WT, 10 Δ*trxl1*, 10 Δ*trxl2*, and 1 Δ*trxl1*Δ*trxl2* parasites, respectively); (iii) detergent treatment with 1.5% cholic acid followed by trypsin digestion (27, 4, and 54 MTs were measured from 10 WT, 1 Δ*trxl1* and 10 Δ*trxl2* strains, respectively). Only visible cortical MTs within the cell bodies were measured. Each data point represents the measured length of a cortical MT, and each color represents one parasite. The numerical data were expressed as mean ± SD (the mean values are indicated in the plot as horizontal lines), ****$P < 0.0001$, two-sided Kruskal–Wallis test with Dunn's Multiple Comparison Correction Test. **b** Representative negative-stain EM images of the whole parasite. The samples were stained with 1% aqueous phosphotungstic acid (PTA). In the image of Δ*trxl1* with cholic acid treatment alone, the posterior ends of visible MTs were marked by yellow asterisks. For Δ*trxl1* with cholic acid treatment plus trypsin digestion, no cortical MTs were observed in most of the cell bodies, while some 'bare' MTs can be occasionally seen in the background, as shown in the red inset. Those bare MTs were not included in the length measurement.

TrxL2 specifically localizes to the cortical MTs (Fig. 5d), indicating that the thioredoxin-like fold itself has little or low affinity for MT lattice. Therefore, it is unclear if MT stabilization is the primary function of TrxL1/2. It is noteworthy that both TrxL1 and TrxL2 lack the canonical CXXC catalytic site[38], suggesting that they are not functional thioredoxin enzymes. So far, only one thioredoxin domain-containing protein (Txl-2) has been reported to interact with MTs in human[39]. Interestingly, Txl-2 is expressed predominantly in the cilia of lung airway epithelium and the manchette and axoneme of spermatids[39,40].

What could be other functions of TrxL1/2? One possibility is that TrxL1/2 serves as adaptors to recruit other proteins, such as tubulin modifying enzymes, to specific locations within the luminal surface of the cortical MT. A previous pull-down experiment revealed a group of TrxL1-interacting proteins including SPM1, TrxL2, and TLAP1/2/3/4[17]. Some of these TLAP proteins could be recruited to the MT lumen via TrxL1 and play certain functions. In our cryo-EM structure, we could not observe additional protein densities near TrxL1/2, even at a very low isosurface threshold (Supplementary Fig. 3) or after focused 3D classification. However, if TLAP proteins do not have strict longitudinal periodicity, their densities would be averaged out in our 3D reconstruction (see further explanation in "Methods").

Another possibility is that the MT-stabilization effect by the MIPs is more important at specific stages of the parasite's life cycle, which may not be represented by our in vitro tissue cultivation condition that only captures the rapidly replicating tachyzoites stage. This may explain the lack of obvious growth phenotypes upon deletion of MIP genes in our experiments and others[16–18]. In the life cycle of *T. gondii*, the MT-stabilization effect provided by certain MIPs could be essential for motility and morphogenesis during acute infection or in response to immune pressure that accompanies chronic infection, or during other stages of their life cycle such as the chronic stages, or during sexual transmission in the cat[41]. Consistent with this idea, *P. falciparum* has different numbers of cortical MTs at different stages (21 for gametocytes and 2–3 for merozoites)[8]. Furthermore, for *T. gondii*, the expression level of TrxL1, TrxL2, and SPM1 vary over different stages (http://ToxoDB.org)[42]. We therefore speculate that the MIP composition and their functions could change at different stages of the life cycle. Future in vivo studies are needed to investigate the MIP functions at different life stages, and across different Apicomplexa species.

Notably, TrxL1 and SPM1 are present in most apicomplexan species, while TrxL2 is not encoded in *Cryptosporidia*, *Piroplasma*, and *Plasmodium*[18], implying that TrxL2 may be functionally redundant to TrxL1. In our cryo-EM structure, it is unclear what factor directs TrxL2 to sites x and y (Fig. 1c). It could be the space restriction and/or orientation requirement near the apex of the ellipse that disfavor the larger TrxL1 molecules containing the N-terminal helix. Another possibility is that

TrxL2 may be guided by the inner membranous particles (IMPs) on the alveoli (see below) or other scaffold proteins that dissociate after finish assembling the cortical MTs.

A previous study has shown that four or more of the six repeat (R) domains of SPM1 are required for proper localization to MTs, while a SPM1 construct lacking the N-terminal domain only localizes to the apex region near the apical polar ring (APR) (Fig. 1a)[16], a non-canonical MTOC that nucleates the cortical MTs[5]. Combined with our structural results, we propose a model for SPM1 aided cortical MT nucleation and elongation. First, SPM1 is recruited to the APR via its C-terminal domain and helps to stabilize the newly formed MTs via its R domains that directly bind to tubulin dimer. The N-terminal domain of SPM1 may interact with the C-terminal domain of another SPM1, and this head-to-tail association (as is the case for FAP363) allows SPM1 to self-propagate along the protofilament and fully expose their R domains, which continue to stabilize the tubulin lattice and facilitate cortical MT elongation. At the same time, SPM1 also recruits other MIPs such as TrxL1/2, which additionally strengthen the MT lattice in the lateral direction (Fig. 3g).

Consistent with this model, the N-terminus of SPM1 in our structure is pointing towards the MT plus end, which in parasites points towards the posterior. It is noteworthy that SPM1 homologs in other Apicomplexa all have the conserved N- and C-terminal domains, while the number of R domains ranges between 5 and 13 copies[16]. We predict that all the SPM1 homologs bind cortical MTs of Apicomplexa using a similar linear arrangement (one R domain per tubulin dimer) as revealed by our cryo-EM structure. Future genetic and structural studies are needed to determine (i) whether SPM1 and its homologs bind cortical MTs using all the tandem R domains in vivo; (ii) whether the N- and C- termini of SPM1 do interact in vivo; (iii) whether the R domains bind all the 13 protofilaments in a coherent register, i.e., the R domains bound at the same "height" (on the same helical turn of the 3-start helix) have the same sequence. In this case, SPM1 would have a true repeat length longer than 8 nm that may set the overall periodicity for the MT lumen.

Previous freeze-fracture experiments[12] have revealed the presence of inner membranous particles (IMPs) on the alveoli that follow the path of the cortical MTs but extend beyond the length of the cortical MTs, suggesting that the placement of cortical MTs might be guided by proteins within the IMC. In addition, bridge-like protein densities between the 27 nm spaced cortical MTs and IMC in *Plasmodium* have been directly observed by cryo-ET[6]. A more recent study also showed that the cortical MTs became disorganized upon deletion of an integral membrane protein GAPM1a within the IMC[43]. All these observations indicate that there are physical interactions between cortical MTs and the membrane (pellicle), presumably via membrane-embedded proteins. In our cryo-EM structure, we did not observe any defined protein densities at the external surface of the cortical MTs, even

at a very low isosurface threshold (Supplementary Fig. 3). This may be due to the limitation of applying single-particle reconstruction to a filamentous object. In the case of MT, any density whose periodicity is not 8 nm or multiples of 8 nm will be averaged out in the reconstructed map and cannot be recovered using 3D classification[25] (see further explanation in "Methods"). Another possibility is that proteins interacting with the external surface of the MTs were disrupted by our sample treatment, which involves membrane removal.

It is also unclear if all the cortical MTs bind to the IMC in the same orientation, using the same protofilament within the asymmetric organization (Fig. 1b, c). A plot of the angular distribution of all the MT particles in our dataset revealed a dominant view (Supplementary Fig. 2c). Considering that the parasites are flattened on the EM grid (Supplementary Fig. 2b), most cortical MTs will be positioned with their IMC-interacting site facing straight up or down. Therefore, the direction of this dominant view suggests that each *T. gondii* cortical MT uses the surface near its seam (or the site opposing the seam, which cannot be distinguished by single-particle analysis) to interact with the IMC. Further in-situ cryo-ET studies of the parasites are needed to confirm this hypothesis.

Since part of the role of the cortical MTs and the associated IMC may be to push against pellicle deformations caused by the action of the glideosome machinery[44], it is satisfying to imagine that the MIPs stabilize MTs from the inside, leaving their surface free to interact with the IMC/pellicle during motility. This seems to be a common scheme for building highly stable MTs that are essential for cell motility, such as the ciliary doublet MTs.

In summary, we have determined the ex vivo structure of singlet MTs with natively decorated proteins from *T. gondii*, an important human parasite that chronically infects around one-third of the world's population. The structure reveals three MIPs (TrxL1, TrxL2, and SPM1), which form a mesh on the luminal surface of the MT and simultaneously stabilize the tubulin lattice in both longitudinal and lateral directions (Fig. 3f). The SPM1 protein shares remarkable similarity to other Mn-motif containing proteins bound to ciliary or cytosolic MTs within distinct species ranging from *Chlamydomonas* to human. We also visualize a small molecule-like density at the Taxol-binding site of β-tubulin that may additionally stabilize the cortical MTs. Our results provide the structural basis to understand the remarkable stability of cortical MTs from *T. gondii* and suggest an evolutionarily conserved mechanism of MT stabilization from the inside.

## Methods

**Parasite and host cell culture.** *T. gondii* tachyzoites were propagated in human foreskin fibroblasts cells (HFFs) at 37 °C, as previously described[45]. The parasites were purified by filtration through 3.0 μm polycarbonate membranes after the natural egress and determined to be mycoplasma negative using the e-Myco plus kit (Intron Biotechnology). RHΔku80Δhxgprt was used as the parental line for CRISPR-Cas9 mediated knockout and tagging.

**CRISPR-Cas9 mediated gene deletion.** To generate single sgRNA-containing CRISPR-Cas9 plasmids targeting 5′UTR and 3′UTR of *trxl1*, *trxl2*, and *spm1* genes, we used pSAG1:CAS9-GFP, U6:sgUPRT[46] as the template to amplify CRISPR-Cas9 vector and performed Gibson assembly using sgRNA-containing primers. Then we constructed double sgRNA-containing CRISPR-Cas9 plasmids using the strategy previously described[47]. Amplicons containing the DHFR resistance cassette flanked by two 40-bp gene-specific 5′ and 3′ homology regions (HR1 and HR2) were generated by PCR using pLoxP-DHFR-TS-mCherry (Addgene plasmid 70147) as the template. 20 μg double sgRNA-containing CRISPR-Cas9 plasmids were electroporated together with 5 μg DHFR amplicons into 5 × 10⁶ freshly harvested parasites, followed by continuous passage under the positive selection of 3 μM pyrimethamine after 24 h's culture in D3 medium at 5% CO₂ and 37 °C. Resistant parasites were then subcloned on HFF monolayers in 96-well plates after examination by diagnostic PCR of the genomic locus. To generate double knockout of *trxl1* and *trxl2* genes, we amplified HXGPRT resistance cassette flanked by 40-bp

gene-specific 5′ and 3′ homology regions of *trxl2* from pTUB1:YFP-mAID-3HA, DHFR-TS:HXGPRT to introduce another selection marker different from DHFR and performed the electroporation together with *trxl2*-targeting CRISPR-Cas9 plasmid into *trxl1* knockout (Δ*trxl1*) which contained DHFR selection marker as described above. Stable transfectants were selected with mycophenolic acid (MPA) (25 μg/ml) and 6-xanthine (6Xa) (50 μg/ml). All the primers are listed in Supplementary Data 2.

**CRISPR-Cas9 mediated gene tagging.** For C-terminal HA tagging, single sgRNA-containing CRISPR-Cas9 plasmids targeting genes of interest locus near the translation stop codon were generated by Gibson assembly. Amplicons containing 3-HA tag, floxed HXGPRT with two 40-bp homology arms were generated by PCR using pTUB1:YFP-mAID-3HA, DHFR-TS:HXGPRT[45] as the template. CRISPR-Cas9 plasmids were co-transfected with amplicons followed by selection with mycophenolic acid (25 μg/ml) and 6-xanthine (6Xa) (50 μg/ml). HA-tags were confirmed by immunofluorescence assay (IFA). All the primers are listed in Supplementary Data 2.

**Plaque assay.** Freshly harvested parasites were counted, and 200 parasites were added to 6-well plates of confluent HFF monolayers in D3 medium. The plaques were then given 8 days to develop. Plaque formation was assessed by counting the zones of clearance on EtOH-fixed, crystal violet-stained HFF monolayers.

**Western blotting.** Parasites protein samples were prepared in X5 Laemmli buffer containing 100 mM dithiothreitol, boiled for 5 min, separated on polyacrylamide gels by SDS-PAGE (Bio-Rad Laboratories, Inc.), and transferred to nitrocellulose membrane. Membranes were blocked with 5% nonfat milk, then probed with rabbit anti-HA (1:1000 dilution, BioLegend) and rabbit anti-MIC2 antibodies (1:1000 dilution, prepared in the Sibley lab[48]) in blocking buffer . Membranes were washed with PBS + 0.1% Tween 20, then incubated with IRDye® 800CW Goat anti-rabbit IgG (1:10,000 dilution, LI-COR Biosciences) and IRDye® 680RD Goat anti-rabbit IgG (1:10,000 dilution, LI-COR Biosciences) in blocking buffer. Membranes were washed several times before scanning on a LiCor Odyssey imaging system (LI-COR Biosciences).

**Immunofluorescence imaging.** *T. gondii* infected HFF monolayer on glass coverslips were fixed in 4% (v/v) formaldehyde in PBS for 15 min and permeabilized with 0.25% (v/v) Triton X-100 (TX-100) in PBS for 20 min at room temperature. Cells were blocked with 3% (w/v) bovine serum albumin (BSA) in PBS overnight at 4 °C. Antibodies for immunofluorescence assay (IFA) include rabbit anti-HA (BioLegend) and mouse anti-acetylated Tubulin (Sigma), goat anti-rabbit IgG Alexa Fluor 488 (Thermo Fisher Scientific), and goat anti-mouse IgG Alexa Fluor 568 (Thermo Fisher Scientific). All antibodies were diluted 1:1500 in the blocking buffer. Coverslips were sealed onto slides with ProLong™ Gold Antifade Mountant with DAPI (Thermo Fisher Scientific). Slides were analyzed using a Zeiss LSM880 laser scanning confocal microscope with Airyscan detection and processing. High-resolution images with an optical Z slice of 0.21 μm were obtained using a ×63, 1.4 numerical aperture Zeiss Plan Apochromat oil objective and ZEN 2.1 black edition software (Carl Zeiss Inc).

**Glycerol extraction and Triton X-100 treatment.** Two T175 flasks of parasites (a total of ~2 × 10⁹ parasites) were harvested soon after egress from the host cells and stimulated to protrude the conoid by treatment with 5 μM A23187 in 5 ml PBS/CaCl₂ buffer (PBS + 4 mM CaCl₂) for 10 min at 37 °C. The stimulated parasites were harvested and resuspended in 5 ml cyto/glycerol buffer (50 mM Tris pH 7.0, 30 mM KCl, 2 mM MgCl₂, 2 mM EGTA, 50% glycerol). The cells were agitated at room temperature for 1 h and centrifuged at 1500 × g for 10 min. The pellet was resuspended in 2 ml cyto/TX-100 buffer (50 mM Tris pH 7.0, 30 mM KCl, 2 mM MgCl₂, 2 mM EGTA, 0.6% TX-100) and incubated at room temperature for 30 min, followed by centrifugation at 1500 × g for 10 min. The pellet was washed with 500 μl cyto buffer (50 mM Tris pH 7.0, 30 mM KCl, 2 mM MgCl₂, 2 mM EGTA), centrifuged again and resuspended in 15 μl cyto buffer.

**Detergent extraction.** Eight T175 flasks of parasites (a total of ~8 × 10⁹ parasites) were harvested for detergent extraction and treated by A23187. The stimulated parasites were centrifuged at 800 × g for 10 min and the supernatant was discarded. The cell pellet was resuspended in 32 ml extraction buffer (10 mM Tris pH 7.4, 50 mM KCl, 1.5% cholic acid (Anatrace Inc)) and incubated at room temperature for 50 min until the solution changed from cloudy to semi-transparent. Then the sample was centrifugated again at 1200 × g for 10 min. The white pellet was resuspended in 50–60 μl lysis/DNase buffer (10 mM Tris pH 7.4, 50 mM KCl, 0.3 μg/μl DNase (Sigma)), incubated on ice for 15 min, centrifuged and resuspended in 50–60 μl lysis buffer (10 mM Tris pH 7.4, 50 mM KCl). The parasite samples after detergent extraction were used for negative stain EM and cryo-EM.

**Trypsin digestion.** Cholic acid extracted *T. gondii* cells (600 μg) were digested by 0.01 mg/ml Trypsin in 20 μl total volume on ice for 45 min. After centrifugation at 1000 × g at 4 °C for 10 min, the pellet was resuspended in 1 ml extraction buffer

(10 mM Tris pH 7.4, 50 mM KCl, 1.5% cholic acid). The centrifugation/resuspension process was repeated three times. The pellet from the final centrifugation was resuspended in 1 ml extraction buffer and incubated on ice for 30 min, then centrifuged and resuspended in 10 μl lysis buffer (10 mM Tris pH 7.4, 50 mM KCl). The parasite samples after trypsin digestion were used for Mass-spectrometry (M/S) analysis.

**Negative-stain electron microscopy.** Samples for negative-stain EM were prepared and imaged using two different protocols. (i) In the Zhang lab, 4 μl of parasite sample was applied onto a glow discharged 400 mesh copper grid coated with a layer of continuous thin-carbon film (Ted Pella, Inc.). After 30 s of adsorption, the sample side of the grid was mixed three times with a drop (30 μl) of 2% uranyl acetate solution laid on parafilm, each time for 30 s. The grid was then blotted with Whatman filter paper to remove the excess sample, air-dried at room temperature, and examined with a JEOL JEM-1400 120 keV microscope (JEOL USA) equipped with an LaB6 filament. Images were recorded using an AMT XR111 high-speed 4k × 2k pixel CCD camera (Advanced Microscopy Techniques), with a defocus range of −1 to −2 μm. (ii) In the Sibley lab, grids were stained with 1% aqueous phosphotungstic acid (PTA), pH 7 (Electron Microscopy Services) for 30 s, air-dried at room temperature, and examined with a JEOL 1200EX 120 keV microscope (JEOL USA) equipped with an AMT 8 mega-pixel CCD camera. The negative-stain EM images from the Zhang lab were mainly used for sample screening, while the images from the Sibley lab were used for final publication (Fig. 6 and Supplementary Fig. 9).

**Cortical MT length measurement.** All the negative-stain images used for MT length measurements were taken in the Sibley lab at the same magnification (×10,000) using the same microscope (JEOL 1200EX TEM). Measurements were done using ImageJ. Only visible cortical MTs within the cell bodies were considered. 'Bare' MTs outside the cell bodies, such as those observed for the Δtrxl1 strain with cholic acid treatment plus trypsin digestion, were not measured. 27–68 visible cortical MTs from 8 to 10 parasites were measured for each condition, except for those with MT length close to 0. Numerical data were expressed as the mean ± SEM and were compared by Ordinary one-way ANOVA or Kruskal–Wallis test. Differences in values were considered significant at ****$P < 0.0001$.

**Mass spectrometry (M/S) analysis.** The _T. gondii_ sample was treated with 1.5% cholic acid and trypsin to remove as much cellular material as possible. Two replicates of the treated sample in the form of frozen pellet were sent for M/S analysis at the Proteomics and Metabolomics Facility at the University of Nebraska-Lincoln. All MS/MS samples were analyzed using Mascot (Matrix Science, version 2.6.1). Mascot was set up to search the ToxoDB-28_TgondiiME49_Annotated Proteins 20160816 database (8322 entries) and the common contaminants database cRAP_20150130 (117 entries).

A total of 608 proteins were identified in the sample. α,β-tubulin and all the three MIPs we identified in this study were at the top of the protein list ranked by the exponentially modified Protein Abundance Index (emPAI), which is a method of estimating protein abundances from peptide counts[49]. The annotated mass spectrometry data are provided as Supplementary Data 1.

**Cryo-EM sample preparation and data collection.** Cryo-EM grids were prepared using a Vitrobot Mark IV (Thermo Fisher Scientific) operated at 16 °C and 95% humidity. 3.5 μl of detergent (cholic acid) extracted parasite sample was applied to a glow discharged Quantifoil R2/2 300 mesh copper grid. After blotting for 4 s with a blot force of −15, the grids were plunge frozen in liquid ethane.

Cryo-EM data were collected over two sessions using a 300 keV Titan Krios microscope equipped with a Cs-corrector (Thermo Fisher Scientific) and a Bioquantum Energy Filter (slit width 20 eV) (Gatan) at the Washington University Center for Cellular Imaging (WUCCI). All data were collected using a K2 Summit direct electron detector (Gatan) operated in counting mode, with an exposure rate of 8.5 electrons/pixel/s on the detector camera. The images were recorded at a defocus range of −1.0 to −3.5 μm with a nominal magnification of ×105,000, which corresponds to a calibrated pixel size of 1.096 Å. A total exposure time of 9 s, corresponding to a total dose of 63.7 electrons/Å² on the specimen, was fractionated into 30 movie frames. EPU software (Thermo Fisher Scientific) was used for semi-automated data collection.

**Cryo-EM data processing.** A total of 9231 movie stacks were motion-corrected and electron-dose weighted using the MotionCor2 program[50]. 4558 micrographs containing visible cortical MTs were manually selected for further processing. The contrast transfer function (CTF) parameters for each motion-corrected micrograph were estimated using Gctf[51]. Following an established protocol for processing MT data[33], cortical MTs were manually picked from the motion-corrected micrographs using the APPION image-processing suite[52]. The selected MT segments were computationally cut into overlapping boxes (512 × 512 pixels) with an 8-nm non-overlapping region (step size) between adjacent boxes (corresponding to the length of a tubulin heterodimer). We refer to these boxed images as 8-nm MT particles. In total, 451,204 particles were used for subsequent data processing.

Starting without any prior knowledge of the structure of the _T. gondii_ cortical MT, we used 13- and 14-protofilaments bovine GMPCPP-MTs[33,34] as the initial models, and performed multi-reference sorting of all the 8-nm MT particles using EMAN1[53]. The classification result indicated that _T. Gondii_ cortical MTs are exclusively 13-protofilament MTs. Next, we performed single-reference refinement using Refine3D in RELION-3 software[54] with global search of alignment parameters. The structure gradually converged to a 13-protofilaments MT with 3-start helix and only one seam. In this structure, the MIP densities displayed 8-nm periodicity and an asymmetric arrangement that is very close to the final structure. After a decent 3D structure was obtained, we realigned all the MT particles using EMAN1 (for global search) and FREALIGN v9.11[55] (for local refinement). We also employed a previously developed MT seam search protocol[56] to minimize the number of misaligned particles.

After alignment parameters were established for each 8-nm MT particle using EMAN1 and FREALIGN, we switched back to RELION-3 for subsequent data processing. We first performed refinement using Refine3D, with only local search of alignment parameters. In the next step, local CTF parameters for each 8-nm MT particle were refined using CTF refinement, which improved the resolution of the structure, and clearly displayed two sets of defocus values that are typically 1000-2000 Å apart (Supplementary Fig. 2a), corresponding to the two sets of cortical MTs located at the 'front' and 'back' surfaces of the parasites (Supplementary Fig. 2b). After CTF refinement, Bayesian particle polishing was performed, which improved the resolution of the overall structure to 4.0 Å.

To further improve the resolution for local regions, we employed a divide-and-conquer strategy[25] by focusing on smaller sub-regions containing only two protofilaments (P2-3, P4-5, P6-7, P8-9, P10-11, P11-12, P13-1) using soft-edged wedge masks (Supplementary Fig. 1d). We first performed 3D classification (Class3D in RELION-3) of different sub-regions using wedge masks with longer length (400 Å) (Supplementary Fig. 1e), while keeping the alignment parameters for each particle from the previous Refine3D run. The Class3D runs typically revealed a dominant subclass with well-defined structural features. Subsequent Refine3D runs for the particles from the dominant subclass used a wedge mask with a shorter length (180 Å) further improved the resolution for the local sub-regions to around 3.5 Å (Supplementary Fig. 1d), which was sufficient for model building. Next, we used cryoSPARC local refinement to recalculate the 3D reconstructions for different sub-regions, which have more consistent gray level (and background noise level) compared to the ones from RELION-3. The 3D reconstructions obtained by cryoSPARC[57] were auto-sharpened using a deep-learning-based software DeepEMhancer[58], which produced better quality maps (especially for the TrxL2 densities) compared with the traditional B-factor sharpening.

To separate the particles with and without TrxL2, we signal-subtracted the tubulin densities from the raw particle images using Particle subtraction in RELION-3, and then performed 3D classification with a soft-edged cylindrical mask covering only the TrxL2 densities at either site x or y. For each site, the 3D classification was able to produce two clean classes, corresponding to the bound and unbound states of TrxL2. Therefore, we calculated two separate structures (with and without TrxL2) for the sub-regions P6-7 (containing site x) and P11-12 (containing site y).

The 3D reconstructions for different sub-regions obtained by cryoSPARC (Supplementary Fig. 1d) were aligned to the overall map (the one at 4.0 Å resolution) using fit in map command in Chimera[59] and merged to produce a composite map using the vop maximum command in Chimera. For figure making and to guide model building, we employed the same merging strategy to generate a composite map for local pieces that were sharpened by DeepEMhancer. We also computed the conventional B-factor sharpened composite map for model refinement, using the bfactor program (http://grigoriefflab.janelia.org/bfactor), with low-pass filter frequency of 3.4 Å and B-factor of −65 (as determined by cryoSPARC). The central portion (130 Å in length, with the best quality) of the composite map was extended to be longer filament (300 Å in length) based on the helical symmetry for better visualization and model refinement.

In single-particle analysis of a filamentous object, protein signals with periodicities longer than 8 nm are lost in a 3D reconstruction when 8-nm periodicity is imposed. However, those with a periodicity that is multiple of 8 nm (e.g., 32 nm) can be recovered by doing 3D classification. We have successfully employed this strategy to reveal MIPs with 48 nm periodicity from a 3D reconstruction with 8-nm periodicity imposed (8 nm map)[25]. In fact, one can typically see these densities in the 8 nm map at lower isosurface threshold. However, this strategy requires the protein to have a relatively big size and a globular domain and cannot recover signal for a protein that does not have strict periodicity.

To look for potential longer periodicity than 8 nm for the identified MIPs and other potential MIP/MAP densities such as the reported TLAP proteins[17], we employed the aforementioned 3D classification strategy. We first signal-subtracted the tubulin densities from the raw particle images using Particle subtraction in RELION-3, and then performed a series of 3D classification runs using cylindrical or wedge masks that only cover a portion of the structure. These attempts did not detect any longer periodicity than 8 nm for the identified MIPs or the presence of other MIPs/MAPs.

**Model building and refinement.** TrxL1 molecule was identified using automated density-guided fold recognition. Globular protein densities were cut from the map

and compared with a library of protein domains using the MOLREP-BALBES pipeline[60]. TrxL2 was considered based on its similarity to TrxL1. SPM1 was considered based on its similarity to FAP363 from *Chlamydomonas*[25]. For the top 100 proteins from mass spectroscopy analysis, their secondary structure profiles were predicted by PSIPRED[61] and used to aid protein assignment and model modeling. Ultimately, the assignment of all MIPs was done by the match between protein sequence and well-resolved side-chain densities.

All modeling was performed in Coot[62]. During real-space refinement in Coot, torsion, planar peptide, and Ramachandran restraints were used. The initial models of TrxL1/2 molecules were build based on the best quality densities at sites b and y (Fig. 1c), respectively, using PDB: 4FYU as the template. The initial models of SPM1 and α,β-tubulin were build based on the densities at protofilament P4, using the atomic models of FAP363 (PDB 6U42)[25] and bovine GDP-MT (PDB 3JAS)[33] as the template, respectively. The initial models of TrxL1/2, SPM1, and α,β-tubulin were then fitted as rigid-bodies into the rest of their corresponding densities and adjusted as needed. In this way, a full helical turn of the models were assembled and then refined into the final sharpened composite map from cryoSPARC using Phenix.real_space_refine v1.18.2-3874[63]. Secondary structure, Ramachandran, and rotamer restraints were applied during refinement. The nonbonded_weight value was set to 500 to reduce the clashes between atoms. The quality of the refined model was assessed by MolProbity[64], with statistics reported in Supplementary Table 1.

**Figures**. All the structural figures were generated using Chimera[59] or ChimeraX[65].

**Reporting summary**. Further information on research design is available in the Nature Research Reporting Summary linked to this article.

## Data availability
Two composite DeepEMhancer-sharpened maps (with and without TrxL2) of *T. gondii* cortical MT are deposited in the Electron Microscopy Data Bank (EMDB) with accession code EMD-23869 and EMD-23870. The original unsharpened composite maps from cryoSPARC and B-factor sharpened composite maps (with and without TrxL2) are associated with these depositions as additional files. An atomic models for one full helical turn containing α,β-tubulin, SPM1 and TrxL1/2 are deposited in the wwPDB with accession code 7MIZ. The two databases used in our mass spectrometry (M/S) analysis are publicly available. The ToxoDB-28_TgondiiME49_Annotated Proteins 20160816 database is available at https://toxodb.org, and the common contaminants database cRAP_20150130 is available at https://www.thegpm.org. The source data underlying Figs. 5a–c, 6a, and Supplementary Fig. 8b–e are provided as a Source Data file. All other data are available from the corresponding authors upon reasonable request. Source data are provided with this paper.

## Code availability
Code used to obtain initial alignment parameters of MT particles by EMAN1[53] and FREALIGN[55] is available at https://github.com/rui--zhang/Microtubule.

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

## Acknowledgements

We thank Michael Rau and James Fitzpatrick at the Washington University Center for Cellular Imaging (WUCCI) for cryo-EM support, and Michael Naldrett at the Proteomics and Metabolomics Facility at the University of Nebraska-Lincoln. Research in the Sibley laboratory is supported by funds from NIAID (AI034036). Research in the Zhang laboratory is supported by departmental startup funds.

## Author contributions

X.W. and Y.F. prepared the HA-tagged and knockout strains of individual MIPs. Y.F. cultured the cells and performed the plaque assays. X.W. extracted and treated the parasites under different chemical conditions. X.W. and M.M. prepared the samples for cryo-EM. X.W. and R.Z. collected cryo-EM images and processed the data. X.W., A.B., and R.Z. built the atomic models. X.W. and W.L.B. performed the negative-stain EM experiments. Y.F. and W.L.B. did the immunofluorescence microscopy. L.D.S. and R.Z. supervised the research and wrote the manuscript with input from all authors.

## Competing interests

The authors declare no competing interests.
