## [Peer Review File · Nature Communications]

REVIEWER COMMENTS

Reviewer #1 (Remarks to the Author):

In this paper the authors describe the discovery and structure of three microtubule inner proteins (MIPs) from *Toxoplasma gondii*. MIPs have been suggested to be present in microtubules from *Toxoplasma* and *Plasmodium* and have recently been discovered in flagellar and cytoplasmic microtubules from a wide range of organisms. Their description now is highly interesting for parasitologists and microtubule cell biologists and the structural work is state of the art. Also the presence of a potential small molecule as microtubule stabilizer is highly intriguing. Unfortunately, the exciting discovery and beauty of the MIPs is not awarded with an impressive phenotype upon the deletion of the genes encoding the three discovered MIPs. Surprisingly, the deletions resulted in no growth defects at all. Only after disruptive biochemical treatments could a difference between wild type and knock out parasites been observed suggesting that the MIPs contribute substantially to microtubule stability. This, was essentially already shown for SPM1 in Tran et al., 2012 and the observation that they play a role in co-recruitment was already shown in Liu et al 2013 as were double knockouts. The biological experiments of the authors therefore add little new data and are not informative. However, the authors did not probe multiple gene deletions and a revised manuscript should at least include the investigation of a TrxL2 / SPM1 double knockout. Also, some investigation of whether K40 acetylation impacts MIP localization could be beneficial. Alternatively, the paper could stand by itself with just the structural data. Next to these possible experimental works, the manuscript should much benefit from a major reworking of the text as indicated below.

Necessary and suggested modifications

General: the introductory and discussion text would much benefit from a shift in focus from *Toxoplasma* to Apicomplexa describing for example the different stages in *Plasmodium* and what has been done there on microtubules; the discussion needs to be much expanded.

Title: mention MIPs?

Main page 1 line 9: *P. falciparum* is not THE causative agent of malaria, but one of several

Line 10: delete the word excellent.

Line 16: reference 8 refers to a *Plasmodium* paper

Sentence ending line 22 needs reference

Line 25: not just in *T. gondii* but also in other apicomplexa and trypanosomes

Line 26: which field?

Main page 2, line 5: there are also MIPs identified from other organisms, please add.

Results page 2, lines 23-24 should go into introduction

Results page 5, lines 25-26: is there any evidence from any other organism that produces a small molecule that binds the cytoskeleton?

Figure 5: provide a Western blot showing proper size of the HA tagged proteins

The discussion is extremely weak and way too short and should be completely rewritten and expanded

Discussion, line 13: Different *Plasmodium* species and different stages have different numbers of microtubules, e.g. *P. falciparum* merozoites have 3 – please modify. Expand on whether the absence of a growth phenotype in *T. gondii* might just be due to tissue culture conditions. Speculate if in a different stage of this or other parasites MIPs could play a role. What is known from other publications about stage or species specific function of cytoskeletal proteins?

Discussion, page 2, line 9-11: Failing to visualize MAPS, especially these that connect MTs to IMC. The authors explain as follows: "This is likely due to the limitation of applying single particle reconstruction

to a filamentous object, i.e. anything that does not follow the 8 nm longitudinal repeat of the MT will be averaged out in the reconstructed map.”

However, later in text in several places the authors state that the method used for data processing (RELION-3?) excludes the presence of some other periodicities in MAP/MIP longitudinal distribution “with confidence”.

“The relatively weak intensity of TrxL2 in the structure is likely due to substoichiometric binding as we can rule out periodicities longer than 8 nm (such as 16 nm or 24 nm) using a 3D classification protocol that has been successfully applied to reveal the 48 nm repeat of the MIPs within the ciliary doublet microtubule²³.” and

“...we performed Class3D jobs using tubulin-signal-subtracted particles produced by Particle subtraction in RELION-3. These tests did not detect periodicities longer than 8 nm for any of the MIPs” This appears inconsistent and the authors should correct one of these conflicting statements.

Reviewer #2 (Remarks to the Author):

Summary

This manuscript describes characterization of a complex of three microtubule inner proteins (MIPs) that confer unusual stability to cortical microtubules in the apicomplexan parasite *Toxoplasma gondii*. The researchers used in situ cryoEM with single particle reconstruction to characterize native cortical microtubules. They identify internal densities that correspond to a complex of three proteins: TrxL1, TrxL2 and SPM1 which were previously localized to cortical microtubules and shown to form a complex. The structural organization of these proteins nicely corresponds with previous data indicating that TrxL1 and TrxL2 localization to the cortical microtubules is dependent upon SPM1 which is located between the microtubule inner surface and TrxL1/2. The researchers identify an unexpected density in the Taxol site of beta-tubulin which they propose is an endogenously synthesized ligand that stabilizes the cortical microtubules, providing functional redundancy to the MIP complex. This is a provocative idea, but they need to expand upon their data and analysis to make it sufficiently solid as an observation in the manuscript. Lastly, the authors create knock-out lines for the three MIPs and show that microtubule stability is affected. This data has been provided in prior papers and is of lower value.

Specific corrections and comments:

Abstract

Line 4-5: “Unlike the cytosolic microtubules with dynamic properties, these specialized cortical microtubules are extraordinarily stable and resistant to detergent and cold treatments.” The way this is written suggests that there are dynamic cytosolic microtubules in *Toxoplasma*. This is not the case, so perhaps a better way to phrase this would be: “Unlike the dynamic microtubules found in the cytoplasm of metazoan cells, these specialized cortical microtubules are extraordinarily stable and resistant to detergent and cold treatments.”

Line 10-11: “Deletion of these identified MIPs compromises microtubule stability and integrity under challenges by chemical treatments.” Since previous researchers (Tran et al, 2012 and Liu et al, 2016) have demonstrated that these proteins confer detergent and cold stability, please acknowledge this. “Consistent with previous observations, deletion of these identified MIPs compromises microtubule stability and integrity under challenges by chemical treatments.”

Main

Line 20: "the membrane" – this is confusing to non-parasitologists. Perhaps consider using pellicle or IMC-plasma membrane complex.

Page 8, Line 16: The story of the apparent small molecule that occupies the Taxol site is provocative but need more detailed information and discussion. What is the size of this density and how does this compare with known Taxol site ligands? Please illustrate how it is distinct from the density for Taxol. The Taxol site is quite large, and binds known ligands in quite different ways. Can you predict what sort of molecule it would be most consistent with?

SPM1 homologs in other apicomplexans have between 5-13 repeat domains. It would be worth noting this in the manuscript and discussing how SPM1 homologs of different repeat lengths would be accommodated and whether they would interact differently with TrxL1/2. The conserved SPM1 head and tail domains suggest conserved interactions, perhaps with the TrxL1/2 proteins. Since the KO stability data presented here is consistent with previous published experiments, a more interesting and simple experiment to include would be to test whether a headless and/or tail-less SPM1 construct can recruit TrxL1/2 to the subpellicular microtubules using a complementation assay in the SPM1 knockout line.

TrxL1 pull-down experiments led to the discovery of TrxL2 and the TLAPs which associate with different regions of the cortical microtubules. Can you discern additional densities for these if you average distinct spans of the cortical microtubules that show association with these proteins? If not, it would be worth noting that there are additional proteins in the TrxL1 complex that may also be MIPs with non-uniform distributions.

Also, since part of the role of the cortical microtubules and the associated IMC may be to push against pellicle deformations caused by action of the glideosome machinery, it is satisfying to imagine that the MIPs stabilize microtubules from the inside, leaving their surface free to interact with motors that may stretch the IMC/pellicle out along the cortical microtubules during motility.

Naomi Morrissette

Reviewer #3 (Remarks to the Author):

The manuscript by Wang et al. reports the cryo-EM structure of the cortical microtubules of the parasite *Toxoplasma gondii*. The structure reveals the arrangement of three proteins (TRXL1, TRXL2 and SPM1) that decorate the microtubule lumen that the authors are able to identify based on the quality of the maps, sequence analysis and previous functional data. Deletion of these proteins using CRISPR in *T. gondii* shows that TRXL1 and SPM1 affect the stability of the microtubule cortical array under sensitizing conditions using cholic acid. This is a well-written, beautiful and technically solid study that shows the power of cryo-EM when used creatively to give insights into the complexity of microtubule architectures found in various cells and that has largely been inaccessible to high-resolution structural analysis until now. The arrangement of the luminal proteins shows some similarity to that found in the *Chlamydomonas* ciliary doublet, but also shows features unique to *T. gondii*. This study also shows that the regular arrangement of luminal proteins plays important roles in conferring stability for microtubules other than those that make up the axoneme and it is likely that they will become a common feature of other in situ microtubules structure. This work should be published.

I would like the authors to clarify only one point regarding the unassigned density that they see in their maps and that occupies the same site where taxol binds. I am not clear why they are considering taxol here as a possibility – did their sample preparation involve the use of taxol? Their methods for cryo-EM sample preparation do not indicate that it was used. If they indeed did use taxol, then it could

easily be a distorted taxol density due to low occupancy and not a new small molecule found in *T. gondii*. The likelihood of the former I would say is higher. If indeed they did not use taxol, can they add it to their preparations and see whether they can displace the small molecule?

Reviewer #4 (Remarks to the Author):

The microtubule cytoskeleton forms the backbone of eukaryotic cells. Although dynamic microtubules are critical forming dynamic structures such as mitotic spindles, stable microtubules are also critical for a variety of cellular processes. Diverse types of stable microtubule structures are found in a variety of eukaryotic protozoa play critical roles in their physiology and pathology. *Toxoplasma Gondii* is a well-studied parasitic eukaryotic protozoa, in which stable cortical microtubule network helps form their unique shape and orient their parasitic conoid organelle toward one end of the cell. The stability and organization of these microtubules remains poorly understood.

Wang et al driven by the Zhang, Sibley and Brown laboratories undertake an excellent collaborative effort to uncover the mechanism for stabilization of the cortical microtubules in *Toxoplasma Gondii* using a combination of biochemistry, cryo-electron microscopy, model building followed by genetic validation and analyses. Their work reveals remarkable and novel inner microtubule proteins (MIPs) network which stabilize the 13 protofilament cortical microtubules. The authors identify thioredoxin fold proteins, Trx1L and Trx2 which form an array head to tail assembly which bind internally and laterally across multiple protofilaments stabilizes the microtubule organization from the inside. The extended repeats of the protein, Spm1, which is related to the human microtubule associated protein 6 (MAP6), forms the base of the Trx1 binding interface on interior intra-dimer interface. The authors utilize CRISPER knockouts to show the roles of these various proteins and dependency in stabilizing the microtubule structure. This study demonstrates the unique evolution of MIPs such as Spm1 and Trx1L, for 13 protofilament microtubule stabilization, to form a hierarchy of assemblies that stabilize the microtubule structure internally, critically driving the pathology of the parasite *T. Gondii*.

This an impressive multi-disciplinary work which is novel and exciting. The work addresses an exciting long-term problem. I recommend publication of this work with minor revisions specifically related to enhancing the clarity of presentation in their structures. I do not recommend any additional experiments. However, the authors must present clearer figures (1-4) demonstrating the organization of these remarkable MIPs and their polymerized lateral structures across multiple protofilaments in relation to the seam. I believe an additional figure (with few panels) viewed from the inside of the microtubule showing the lateral pattern of these Trx1L and Trx2 proteins should help the reader understand how Trx1L (potentially Trx2) self-assembly laterally and internally across four to five protofilaments stabilize these microtubules. These should also include repeating patterns of the many repeats in Spm1 proteins should be explored longitudinally or laterally internal fashion. The authors should present a basic final model describing the interfaces of these proteins and their roles in promoting the stability of these *T. Gondii* cortical microtubules.

Reviewer #5 (Remarks to the Author):

The manuscript of Wang et al describes the high resolution structure of cortical microtubules of *Toxoplasma gondii* determined using cryo-EM. This structure allows the authors to identify distinct microtubule inner proteins (MIPs) and to visualise how they might contribute to the remarkable stability of this microtubule population. The authors use their structural findings to investigate the contribution of these MIPs to cortical microtubule stability in parasites and describe a hierarchy of stabilisation effects. The work is largely clearly described and accompanied by clear and informative figures. Their findings will be of great interest to cytoskeleton and parasite communities.

My comments mainly relate to ways in which data description and presentation can be more precise and readability improved:

- 1) The determined structure is not truly in situ because parasite treatment prior to vitrification has removed the majority of cell contents (c.f. for example Schur, Curr Op Struct Biol (2019)). Ex vivo is a more appropriate term.
- 2) The Results section text begins oddly and it is initially hard to understand what was done and with what motivation. I suggest that this paragraph be rewritten so the reader can understand what is being attempted and why.
- 3) Almost no views of the microtubule outside surface are included in the manuscript, and more evidence needs to be provided to support the statement (line 25, p5) "On the other hand, we did not observe any defined protein densities at the external surface of the cortical MTs." What does the structure look like at different thresholds, filters, etc? Were masks applied?
- 4) Relatedly, the authors suggest in the discussion that no external MAP densities were seen because of the image processing procedures used. Isn't it at least equally likely that external MAPs were destabilised by parasite treatment and membrane removal prior to vitrification? This possibility should be discussed.
- 5) Please label alpha- and beta-tubulin in Fig 1c.
- 6) Examples of unique side chain densities should be provided to support the assignment of the densities of TrxL1/2.
- 7) In the legend to Fig. 3d it is stated "We chose the most likely internal repeat R4 for model building". Most likely by what criteria?
- 8) Assignment of the density in the taxol binding site as a small molecule is just one explanation for this density – how did the authors exclude that this was not attributable to protein e.g. an extension of nearby SPM1?
- 9) Please check the scale bars in Fig 5e – if the scale bars are correct, the authors should clearly describe and explain the observation that TrxL2-HA and deltaxk1 TrxL2-HA parasites are a different size of the other strains depicted
- 10) The data presented in Extended Data Fig 2a is interesting and important to show, but the figure is not clearly annotated or explained. It would be useful for non-cryo-EM experts to explicitly state that each dotted line corresponds to an individual microtubule. What exactly do blue and orange correspond to? The statement "From blue color to yellow, the range of defocus values spans 2,075 Å" is not sufficient to understand these data.
- 11) Relating to Extended Data Fig 2c – is the anisotropy of the data reflected in the quality of the structure determination and model building? Any effects on interpretation should be discussed, and evidence provided if quality is not affected.

General responses to the reviewers:

We thank all the reviewers for their constructive comments and suggestions. In the revised manuscript, we have:

- (i) Generated and characterized a double knockout strain $\Delta trx1\Delta trx2$, which displays a stronger phenotype than the single knockout upon chemical treatments.
- (ii) Added immunofluorescence assay (IFA) data of two newly made constructs: (1) $\Delta trx2$, TrxL1-HA, (2) $\Delta spm1$, TrxL2-HA, which revealed the interdependence of the MIPs for localization.
- (iii) Introduced a new step in cryo-EM data processing, which allowed us to separate the bound and unbound states of TrxL2 at sites x and y. This classification step improved the quality of the TrxL2 densities. We have added a new panel in Fig. 2 to show the side chain densities that distinguish TrxL2 from TrxL1.
- (iv) Expanded the Discussion section to speculate on the physiological functions of MIPs and propose a new model for SPM1-aided cortical MT nucleation and elongation.
- (v) Added a schematic (Fig. 3g) to summarize the interactions among TrxL1/2, SPM1 and tubulin, and illustrate that the three MIPs form a mesh on the luminal surface and simultaneously stabilize the tubulin lattice in both longitudinal and lateral directions.

The changes in the revised manuscript are shown in blue.

Please also see our point-by-point responses to the reviewer comments below.

REVIEWER COMMENTS

Reviewer #1 (Remarks to the Author):

In this paper the authors describe the discovery and structure of three microtubule inner proteins (MIPs) from *Toxoplasma gondii*. MIPs have been suggested to be present in microtubules from *Toxoplasma* and *Plasmodium* and have recently been discovered in flagellar and cytoplasmic microtubules from a wide range of organisms. Their description now is highly interesting for parasitologists and microtubule cell biologists and the structural work is state of the art. Also the presence of a potential small molecule as microtubule stabilizer is highly intriguing. Unfortunately, the exciting discovery and beauty of the MIPs is not awarded with an impressive phenotype upon the deletion of the genes encoding the three discovered MIPs. Surprisingly, the deletions resulted in no growth defects at all. Only after disruptive biochemical treatments could a difference between wild type and knock out parasites be observed suggesting that the MIPs contribute substantially to microtubule stability. This, was essentially already shown for SPM1 in Tran et al., 2012 and the observation that they play a role in co-recruitment was already shown in Liu et al 2013 as were double knockouts. The biological experiments of the authors therefore add little new data and are not informative. However, the authors did not probe multiple gene deletions and a revised manuscript should at least include the investigation of a TrxL2 / SPM1 double knockout.

We agree that some of the functional data were repeating previous studies. We prefer to keep them in the manuscript for completeness and for independent validation of the previous studies. In the revision, we added several new experimental results: (i) generation and characterization

of a double knockout strain $\Delta trx1\Delta trx2$, which displayed a stronger phenotype than the single knockout upon chemical treatments; (ii) IFA data of two newly made constructs (1) $\Delta trx2$, TrxL1-HA, (2) $\Delta spm1$, TrxL2-HA, which revealed the interdependence of the MIPs for localization.

The reason we didn't pursue the TrxL2 / SPM1 double knockout is as follows. Upon chemical treatments, SPM1 knockout strain showed severe phenotype (cortical microtubules were completely disrupted), while TrxL2 knockout strain behaved similarly to the wild type. Therefore, we believe the TrxL2 / SPM1 double knockout will show the same phenotype as the SPM1 knockout, which will not be informative.

Also, some investigation of whether K40 acetylation impacts MIP localization could be beneficial. Alternatively, the paper could stand by itself with just the structural data.

Ablation of K40 acetylation induces a severe replication defect (ref. 1): parasites appear to initiate mitosis yet exhibit incomplete or improper nuclear division. Therefore, it would be challenging to work with the deacetylated strain to investigate the effects of K40 acetylation on MIP localization.

On a separate note, in our previous published cryo-EM structure of ciliary doublet microtubule (ref. 2), various MIPs were observed to directly interact with the K40-containing loop of α -tubulin and stabilize its conformation. However, in our cryo-EM structure of *T. gondii* MT, we couldn't visualize the K40-loop that is intrinsically flexible, indicating that this loop is not involved in direct inaction with the MIPs. Of course, we cannot rule out the indirect influence of K40 acetylation on MIP localization.

[1] Varberg, J.M., Padgett, L.R., Arrizabalaga, G., and Sullivan, W.J. (2016). TgATAT-Mediated α -Tubulin Acetylation Is Required for Division of the Protozoan Parasite *Toxoplasma gondii*. *mSphere* 1, 1217.

[2] Ma, M., Stoyanova, M., Rademacher, G., Dutcher, S.K., Brown, A., and Zhang, R. (2019). Structure of the Decorated Ciliary Doublet Microtubule. *Cell* 179, 909–922.e912.

Next to these possible experimental works, the manuscript should much benefit from a major reworking of the text as indicated below.

Necessary and suggested modifications

General: the introductory and discussion text would much benefit from a shift in focus from *Toxoplasma* to Apicomplexa describing for example the different stages in *Plasmodium* and what has been done there on microtubules; the discussion needs to be much expanded.

We have modified the Introduction and expanded the Discussion to shift the focus from *Toxoplasma* to Apicomplexa for broader interest.

Title: mention MIPs?

We have changed the title to "Cryo-EM structure of cortical microtubules from human parasite *Toxoplasma gondii* identifies their microtubule inner proteins".

Main page 1 line 9: *P. falciparum* is not THE causative agent of malaria, but one of several

We have modified the text accordingly.

Line 10: delete the word excellent.

Done.

Line 16: reference 8 refers to a Plasmodium paper

In reference 8, it actually states that “when imaging the apical part of *T. gondii* tachyzoites (data not shown), we also found a distance between the IMC and microtubules of 23 nm.” We previously missed this sentence and now have modified the text accordingly.

Sentence ending line 22 needs reference

We have modified this sentence to be more specific to Apicomplexa (as suggested by reviewer #2) and added a reference.

Line 25: not just in *T. gondii* but also in other apicomplexa and trypanosomes

We have modified the text accordingly and added a reference for trypanosomes.

Line 26: which field?

We have changed the sentence to specifically mention the microtubule field.

Main page 2, line 5: there are also MIPs identified from other organisms, please add.

Done.

Results page 2, lines 23-24 should go into introduction

Done.

Results page 5, lines 25-26: is there any evidence from any other organism that produces a small molecule that binds the cytoskeleton?

We are not aware of any. However, our structural finding could be the beginning of a series of exciting discoveries in the near future.

Figure 5: provide a Western blot showing proper size of the HA tagged proteins

We have added the Western blot as Fig. 5c.

The discussion is extremely weak and way too short and should be completely rewritten and expanded.

We have rewritten and expanded the Discussion.

Discussion, line 13: Different Plasmodium species and different stages have different numbers of microtubules, e.g. *P. falciparum* merozoites have 3 – please modify.

Thanks. We have added it to the first part of the Discussion.

Expand on whether the absence of a growth phenotype in *T. gondii* might just be due to tissue culture conditions. Speculate if in a different stage of this or other parasites MIPs could play a role. What is known from other publications about stage or species specific function of cytoskeletal proteins?

This is an excellent point as our work is focused only on *in vitro* cultivation. We have not performed *in vivo* studies and it is possible that stable MTs are even more important during infection, or during other stages of the life cycle such as the chronic stages, or during sexual transmission in the cat. We have added a statement to the Discussion.

Discussion, page 2, line 9-11: Failing to visualize MAPS, especially these that connect MTs to IMC. The authors explain as follows: “This is likely due to the limitation of applying single

particle reconstruction to a filamentous object, i.e. anything that does not follow the 8 nm longitudinal repeat of the MT will be averaged out in the reconstructed map.”

However, later in text in several places the authors state that the method used for data processing (RELION-3?) excludes the presence of some other periodicities in MAP/MIP longitudinal distribution “with confidence”.

“The relatively weak intensity of TrxL2 in the structure is likely due to substoichiometric binding as we can rule out periodicities longer than 8 nm (such as 16 nm or 24 nm) using a 3D classification protocol that has been successfully applied to reveal the 48 nm repeat of the MIPs within the ciliary doublet microtubule²³.” and

“...we performed Class3D jobs using tubulin-signal-subtracted particles produced by Particle subtraction in RELION-3. These tests did not detect periodicities longer than 8 nm for any of the MIPs“ This appears inconsistent and the authors should correct one of these conflicting statements.

We apologize for the confusion. We have modified the relevant statements to make it clearer and added the following statements to the Methods section.

In single particle analysis of a filamentous object, protein signals with periodicities longer than 8 nm are lost in a 3D reconstruction when 8-nm periodicity is imposed. However, those with a periodicity that is multiple of 8 nm (e.g., 32 nm) can be recovered by doing 3D classification. We have successfully employed this strategy to reveal MIPs with 48-nm periodicity from a 3D reconstruction with 8-nm periodicity imposed (Ma et al. 2019). However, this strategy requires the protein to have a relatively big size and a globular domain and cannot recover signal for a protein that doesn't have strict periodicity.

Reviewer #2 (Remarks to the Author):

Summary

This manuscript describes characterization of a complex of three microtubule inner proteins (MIPs) that confer unusual stability to cortical microtubules in the apicomplexan parasite *Toxoplasma gondii*. The researchers used in situ cryoEM with single particle reconstruction to characterize native cortical microtubules. They identify internal densities that correspond to a complex of three proteins: TrxL1, TrxL2 and SPM1 which were previously localized to cortical microtubules and shown to form a complex. The structural organization of these proteins nicely corresponds with previous data indicating that TrxL1 and TrxL2 localization to the cortical microtubules is dependent upon SPM1 which is located between the microtubule inner surface and TrxL1/2. The researchers identify an unexpected density in the Taxol site of beta-tubulin which they propose is an endogenously synthesized ligand that stabilizes the cortical microtubules, providing functional redundancy to the MIP complex. This is a provocative idea, but they need to expand upon their data and analysis to make it sufficiently solid as an observation in the manuscript. Lastly, the authors create knock-out lines for the three MIPs and show that microtubule stability is affected. This data has been provided in prior papers and is of lower value.

Please see our response to reviewer #1. In the revision we have added several new experimental results: (i) generation and characterization of a double knockout strain $\Delta trx1\Delta trx2$, which displayed stronger phenotype than the single knockout upon chemical treatments; (ii) IFA data of two newly made constructs (1) $\Delta trx2$, TrxL1-HA, (2) $\Delta spm1$, TrxL2-HA, which revealed interesting dependencies among MIPs.

Specific corrections and comments:

Abstract

Line 4-5: “Unlike the cytosolic microtubules with dynamic properties, these specialized cortical microtubules are extraordinarily stable and resistant to detergent and cold treatments.” The way this is written suggests that there are dynamic cytosolic microtubules in *Toxoplasma*. This is not the case, so perhaps a better way to phrase this would be: “Unlike the dynamic microtubules found in the cytoplasm of metazoan cells, these specialized cortical microtubules are extraordinarily stable and resistant to detergent and cold treatments.”

We thank the reviewer for the great suggestion and have modified the Abstract and text accordingly.

Line 10-11: “Deletion of these identified MIPs compromises microtubule stability and integrity under challenges by chemical treatments.” Since previous researchers (Tran et al, 2012 and Liu et al, 2016) have demonstrated that these proteins confer detergent and cold stability, please acknowledge this. “Consistent with previous observations, deletion of these identified MIPs compromises microtubule stability and integrity under challenges by chemical treatments.”

Done.

Main

Line 20: “the membrane” – this is confusing to non-parasitologists. Perhaps consider using pellicle or IMC-plasma membrane complex.

Done.

Page 8, Line 16: The story of the apparent small molecule that occupies the Taxol site is provocative but need more detailed information and discussion. What is the size of this density and how does this compare with known Taxol site ligands? Please illustrate how it is distinct from the density for Taxol. The Taxol site is quite large, and binds known ligands in quite different ways. Can you predict what sort of molecule it would be most consistent with?

We have added more panels to Fig. 4 to compare the size and shape of this small molecule with Taxol and other commonly used Taxol site drugs (zampanolide and epothilone). Unfortunately, at this stage we cannot predict the chemical property of this small molecule, even after consulting an expert (J. Fernando Díaz).

SPM1 homologs in other apicomplexans have between 5-13 repeat domains. It would be worth noting this in the manuscript and discussing how SPM1 homologs of different repeat lengths would be accommodated and whether they would interact differently with TrxL1/2. The conserved SPM1 head and tail domains suggest conserved interactions, perhaps with the TrxL1/2 proteins. Since the KO stability data presented here is consistent with previous published experiments, a more interesting and simple experiment to include would be to test whether a headless and/or tail-less SPM1 construct can recruit TrxL1/2 to the subpellicular microtubules using a complementation assay in the SPM1 knockout line.

We have added a section in the Discussion on how SPM1 homologs of different repeat lengths can be accommodated and proposed a model for SPM1 aided cortical MT nucleation and elongation.

In our cryo-EM structure, we observe direct interaction between SPM1 R repeat domain and TrxL1/2. We believe SPM1 head and tail domains do not directly interact with TrxL1/2 (therefore the suggested experiment will not be very informative), otherwise we should be able visualize additional densities near TrxL1/2 at lower isosurface threshold. In the expanded Discussion, we propose that the head and tail domains of SPM1 are involved in head-to-tail self-association of SPM1 along the protofilament (as is the case for FAP363), and the tail domain (C-terminus) of SPM1 interacts with the apical polar ring.

To further investigate the interaction between SPM1 and TrxL2, we have added new IFA data of $\Delta spm1$, TrxL2-HA, which showed that the localization of TrxL2 partially depends on SPM1.

TrxL1 pull-down experiments led to the discovery of TrxL2 and the TLAPs which associate with different regions of the cortical microtubules. Can you discern additional densities for these if you average distinct spans of the cortical microtubules that show association with these proteins? If not, it would be worth noting that there are additional proteins in the TrxL1 complex that may also be MIPs with non-uniform distributions.

The field of view of our cryo-EM images is quite small compared to the dimension of the whole parasite (see Extended Data Fig. 1). As a result, for most of the cryo-EM images, it is difficult to tell which span of the cortical MT they come.

On the other hand, we are confident that if there is any additional density in the lumen with regular periodicity that is multiple of 8-nm, we should be able to detect it using 3D classification. In fact, one can typically see these densities in the 8-nm reconstruction at much lower isosurface threshold. We tried both methods but couldn't see any additional densities near TrxL1/2. We have added the statements to the Methods (please also see our response to reviewer #1 regarding the periodicity).

We agree there could be additional MIPs with non-uniform distributions (therefore are invisible in our cryo-EM structure). We have added one paragraph about potential TrxL1-interacting proteins within the MT lumen.

Also, since part of the role of the cortical microtubules and the associated IMC may be to push against pellicle deformations caused by action of the glideosome machinery, it is satisfying to imagine that the MIPs stabilize microtubules from the inside, leaving their surface free to interact with motors that may stretch the IMC/pellicle out along the cortical microtubules during motility.

We thank the reviewer for the great insights. We have incorporated it to our discussion.

Reviewer #3 (Remarks to the Author):

The manuscript by Wang et al. reports the cryo-EM structure of the cortical microtubules of the parasite *Toxoplasma gondii*. The structure reveals the arrangement of three proteins (TRXL1, TRXL2 and SPM1) that decorate the microtubule lumen that the authors are able to identify based on the quality of the maps, sequence analysis and previous functional data. Deletion of these proteins using CRISPR in *T. gondii* shows that TRXL1 and SPM1 affect the stability of the microtubule cortical array under sensitizing conditions using cholic acid. This is a well-written, beautiful and technically solid study that shows the power of cryo-EM when used creatively to give insights into the complexity of microtubule architectures found in various cells and that has largely been inaccessible to high-resolution structural analysis until now. The arrangement of the luminal proteins shows some similarity to that found in the *Chlamydomonas* ciliary doublet, but also shows features unique to *T. gondii*.

This study also shows that the regular arrangement of luminal proteins plays important roles in conferring stability for microtubules other than those that make up the axoneme and it is likely that they will become a common feature of other in situ microtubules structure. This work should be published.

I would like the authors to clarify only one point regarding the unassigned density that they see in their maps and that occupies the same site where taxol binds. I am not clear why they are considering taxol here as a possibility – did their sample preparation involve the use of taxol? Their methods for cryo-EM sample preparation do not indicate that it was used. If they indeed did use taxol, then it could easily be a distorted taxol density due to low occupancy and not a new small molecule found in *T. gondii*. The likelihood of the former I would say is higher. If indeed they did not use taxol, can they add it to their preparations and see whether they can displace the small molecule?

Taxol was not used throughout our parasite growth and sample preparation process. We have modified the text to make this clear. We have also added more panels to Fig. 4 to show the difference between the observed density and three taxane drugs (Taxol, zampanolide and epothilone).

A previous study (ref. 3) has shown that cortical MTs in *T. gondii* can be labelled by a fluorescent Taxol derivative (named SiR-tubulin, ref. 4) that presumably binds to the same site as Taxol, indicating that this putative endogenous small molecule can be at least partially displaced by Taxol. In the future, we plan to use a Taxol displacement strategy followed by mass spec analysis to aid the identification of this endogenous small molecule. However, this is a non-trivial task and will be a separate paper.

[3] Del Rosario, M., Periz, J., Pavlou, G., Lyth, O., Latorre-Barragan, F., Das, S., Pall, G.S., Stortz, J.F., Lemgruber, L., Whitelaw, J.A., et al. (2019). Apicomplexan F-actin is required for efficient nuclear entry during host cell invasion. *EMBO Reports* 20, e48896.

[4] Lukinavičius, G., Reymond, L., D'Este, E., Masharina, A., Göttfert, F., Ta, H., Güther, A., Fournier, M., Rizzo, S., Waldmann, H., et al. (2014). Fluorogenic probes for live-cell imaging of the cytoskeleton. *Nat Methods* 11, 731–733.

Reviewer #4 (Remarks to the Author):

The microtubule cytoskeleton forms the backbone of eukaryotic cells. Although dynamic microtubules are critical forming dynamic structures such as mitotic spindles, stable microtubules are also critical for a variety of cellular processes. Diverse types of stable microtubule structures are found in a variety eukaryotic protozoa play critical roles in their physiology and pathology. *Toxoplasma Gondii* is a well-studied parasitic eukaryotic protozoa, in which stable cortical microtubule network helps form their unique shape and orient their parasitic conoid organelle toward one end of the cell. The stability and organization of these microtubules remains poorly understood.

Wang et al driven by the Zhang, Sibley and Brown laboratories undertake an excellent collaborative effort to uncover the mechanism for stabilization of the cortical microtubules in *Toxoplasma Gondii* using a combination of biochemistry, cryo-electron microscopy, model building followed by genetic validation and analyses. Their work reveals remarkable and novel inner microtubule proteins (MIPs) network which stabilize the 13 protofilament cortical microtubules. The authors identify thioredoxin fold proteins, Trx1L and Trx2 which form an array head to tail assembly which bind internally and laterally across multiple protofilaments stabilizes

the microtubule organization from the inside. The extended repeats of the protein, Spm1, which is related to the human microtubule associated protein 6 (MAP6), forms the base of the Trx1 binding interface on interior intra-dimer interface. The authors utilize CRISPER knockouts to show the roles of these various proteins and dependency in stabilizing the microtubule structure. This study demonstrates the unique evolution of MIPs such as Spm1 and Trx1L, for 13 protofilament microtubule stabilization, to form a hierarchy of assemblies that stabilize the microtubule structure internally, critically driving the pathology of the parasite *T. Gondii*.

This an impressive multi-disciplinary work which is novel and exciting. The work addresses an exciting long-term problem. I recommend publication of this work with minor revisions specifically related to enhancing the clarity of presentation in their structures. I do not recommend any additional experiments. However, the authors must present clearer figures (1-4) demonstrating the organization of these remarkable MIPs and their polymerized lateral structures across multiple protofilaments in relation to the seam. I believe an additional figure (with few panels) viewed from the inside of the microtubule showing the lateral pattern of these TrX1L and TrX2 proteins should help the reader understand how TrX1L (potentially TrX2) self-assembly laterally and internally across four to five protofilaments stabilize these microtubules. These should also include repeating patterns of the many repeats in Spm1 proteins should be explored longitudinally or laterally internal fashion.

We thank the reviewer for their enthusiasm about our work. We have modified Figs. 1-4 to better illustrate the intricate interactions among these MIPs. We also added a new Extended Data Fig. 4 to show multiple protofilaments across the seam. The possible linear arrangement of SPM1 (in the longitudinal direction) is shown in Extended Data Fig. 5. Based on the continuous SPM1 density in the new panel Fig. 3f, it is impossible for one SPM1 molecule to span across multiple protofilaments in the lateral direction.

The authors should present a basic final model describing the interfaces of these proteins and their roles in promoting the stability of these *T. Gondii* cortical microtubules.

We thank the reviewer for the suggestion. We have added a schematic diagram (Fig. 3g) to summarize the interactions among TrXL1/2, SPM1 and tubulin. We also added two new sections in the Discussion to speculate on the physiological functions of the MIPs and to propose a model for SPM1-aided cortical MT nucleation and elongation.

Reviewer #5 (Remarks to the Author):

The manuscript of Wang et al describes the high resolution structure of cortical microtubules of *Toxoplasma gondii* determined using cryo-EM. This structure allows the authors to identify distinct microtubule inner proteins (MIPs) and to visualise how they might contribute to the remarkable stability of this microtubule population. The authors use their structural findings to investigate the contribution of these MIPs to cortical microtubule stability in parasites and describe a hierarchy of stabilisation effects. The work is largely clearly described and accompanied by clear and informative figures. Their findings will be of great interest to cytoskeleton and parasite communities.

My comments mainly relate to ways in which data description and presentation can be more precise and readability improved:

1) The determined structure is not truly *in situ* because parasite treatment prior to vitrification has removed the majority of cell contents (c.f. for example Schur, *Curr Op Struct Biol* (2019)). *Ex vivo* is a more appropriate term.

We agree with the reviewer. We have changed the term "*in situ*" to "*ex vivo*".

2) The Results section text begins oddly and it is initially hard to understand what was done and with what motivation. I suggest that this paragraph be rewritten so the reader can understand what is being attempted and why.

We have modified the beginning of the Results section to better explain the rationale behind the experimental approach.

3) Almost no views of the microtubule outside surface are included in the manuscript, and more evidence needs to be provided to support the statement (line 25, p5) "On the other hand, we did not observe any defined protein densities at the external surface of the cortical MTs." What does the structure look like at different thresholds, filters, etc? Were masks applied?

We have added a new Extended Fig. 3 to show the external surface (cross-sectional and longitudinal views) of our cryo-EM structure at low isosurface threshold without a mask applied. We also examined the structure filtered to lower resolution, e.g., 20 Å or 30 Å, but couldn't observe any additional densities on the external surface.

4) Relatedly, the authors suggest in the discussion that no external MAP densities were seen because of the image processing procedures used. Isn't it at least equally likely that external MAPs were destabilised by parasite treatment and membrane removal prior to vitrification? This possibility should be discussed.

We have modified the text to acknowledge this possibility.

5) Please label alpha- and beta-tubulin in Fig 1c.

Done.

6) Examples of unique side chain densities should be provided to support the assignment of the densities of TrxL1/2.

We have modified Fig. 2 to include three side chain densities (TrxL2 residues V52, G142 and R143) that clearly distinguish TrxL1/2, in addition to their apparent differences at the N-termini (black arrows in Fig. 2a).

7) In the legend to Fig. 3d it is stated "We chose the most likely internal repeat R4 for model building". Most likely by what criteria?

Sorry, R4 is an arbitrary choice; we are almost certain that the SPM1 density is a mixture of different R repeat domains. We have clarified the choice in the figure legend.

8) Assignment of the density in the taxol binding site as a small molecule is just one explanation for this density – how did the authors exclude that this was not attributable to protein e.g. an extension of nearby SPM1?

We can exclude the possibility of this density being part of nearby SPM1 for two reasons. (i) In our cryo-EM structure, SPM1 is not present on protofilaments P1 and P7 (without TrxL2), but the small densities are, with an intensity level as strong as tubulin. (ii) SPM1 forms a continuous or near-continuous density along each protofilament. Based on our modeling, all the 32 residues of the R domain (Fig. 3e) are needed to fill the space of the continuous density and make a

linear arrangement. Therefore, residues within the R domain cannot turn sideways into the taxol binding pocket.

It is also spatially impossible for the density to be missing residues of TrxL1/2. We have added these statements in the text.

9) Please check the scale bars in Fig 5e – if the scale bars are correct, the authors should clearly describe and explain the observation that TrxL2-HA and *deltatrsk1* TrxL2-HA parasites are a different size of the other strains depicted.

We thank the reviewer for catching this error. Indeed, some of the figures were further zoomed in to show the details better, but the scale bars were not updated accordingly. We have now fixed this error in the revision (note the length of the 2 μm scale bar is slightly different for each sub-panel). We believe there is no significant size difference among different strains.

10) The data presented in Extended Data Fig 2a is interesting and important to show, but the figure is not clearly annotated or explained. It would be useful for non-cryo-EM experts to explicitly state that each dotted line corresponds to an individual microtubule. What exactly do blue and orange correspond to? The statement “From blue color to yellow, the range of defocus values spans 2,075 Å” is not sufficient to understand these data.

We thank the reviewer for the good suggestions. We have added a histogram of “number of particles versus their defocus values” and a color scale bar to Extended Data Fig. 2a and modified the figure legend to make it clearer. The blue / orange color corresponds to the lowest / highest per-particle defocus value.

11) Relating to Extended Data Fig 2c – is the anisotropy of the data reflected in the quality of the structure determination and model building? Any effects on interpretation should be discussed, and evidence provided if quality is not affected.

The uneven angular distribution doesn't seem to impact the quality of the 3D reconstruction, based on visual inspection as well as the similar resolution number for different sub-regions (new Extended Data Fig. 1d). This is probably due to the fact that we have a large dataset and therefore sufficient number of particles for the non-dominant views. We have noted this in the figure legend.

REVIEWERS' COMMENTS

Reviewer #1 (Remarks to the Author):

The manuscript has much improved and the authors adequately addressed my comments. Congratulations to a truly beautiful piece of work that not only discovers new biology but also opens up a path for future exciting discoveries.

Reviewer #2 (Remarks to the Author):

Upon re-review, I am happy with the revisions to this manuscript. I suggest only a small addition: page 1 of the main text -- when discussing the Mt spacing in plasmodium state the life cycle stage (sporozoites) since the MT organization can be quite different in these different forms. Otherwise, I am quite happy with how the comments have been addressed.

Reviewer #3 (Remarks to the Author):

This manuscript should be published as is. It is an important and beautiful study.

Reviewer #4 (Remarks to the Author):

The authors have revised the presentation and added additional experiments that fully address the concerns of all reviewers. The work is suitable for publication.

Reviewer #5 (Remarks to the Author):

I am satisfied with this revised version of the manuscript. Congratulations to the authors on this very nice piece of work.

General responses to the reviewers:

We thank all the reviewers for their positive comments and suggestions.

REVIEWERS' COMMENTS

Reviewer #1 (Remarks to the Author):

The manuscript has much improved and the authors adequately addressed my comments. Congratulations to a truly beautiful piece of work that not only discovers new biology but also opens up a path for future exciting discoveries.

Reviewer #2 (Remarks to the Author):

Upon re-review, I am happy with the revisions to this manuscript. I suggest only a small addition: page 1 of the main text -- when discussing the Mt spacing in plasmodium state the life cycle stage (sporozoites) since the MT organization can be quite different in these different forms. Otherwise, I am quite happy with how the comments have been addressed.

Done.

Reviewer #3 (Remarks to the Author):

This manuscript should be published as is. It is an important and beautiful study.

Reviewer #4 (Remarks to the Author):

The authors have revised the presentation and added additional experiments that fully address the concerns of all reviewers. The work is suitable for publication.

Reviewer #5 (Remarks to the Author):

I am satisfied with this revised version of the manuscript. Congratulations to the authors on this very nice piece of work.